# On the Power of Statistics in Class-Incremental Learning with Pretrained Models

**Zhiwen Cao** [1 2]  **Yanfeng Li** [3]  **Shu-Dong Huang** [1 2]  **Yalan Ye** [4]  **Shuyin Xia** [5 6]  **Yi Wang** [7]  **Jiancheng Lv** [1 2]

## Abstract

Recent class-incremental learning (CIL) methods built on large pre-trained vision models have shown that strong performance can be retained even under strict data access constraints. This raises a fundamental question: which properties of pre-trained representations make such recovery possible in the class-incremental setting? In this work, we show that class-level feature statistics play a central role in enabling effective CIL under strong pre-training. When the visual backbone is frozen, maintaining simple class-wise statistical estimators of features can recover a substantial fraction of the performance achieved by static joint training across diverse benchmarks. We make this observation explicit through deliberately minimal reference points built on frozen CLIP representations. In particular, we demonstrate that competitive performance can be achieved without continual parameter updates, by performing class-incremental inference based solely on class-level statistical estimators instantiated from frozen features. Our findings suggest that class-level statistics constitute an important and previously underemphasized component of recent PTM-based CIL approaches, offering a complementary perspective for understanding their strong empirical performance. Our code is available at https://github.com/HdTgon/baseCIL.

## 1. Introduction

Modern deep neural networks have achieved remarkable success across a wide range of real-world applications. Despite this progress, most models are trained under the assumption

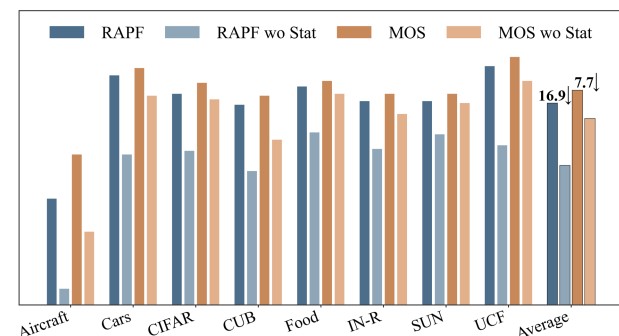

*Figure 1.* Pilot study illustrating the impact of statistics-driven feature replay. Disabling synthetic feature replay based on class-level statistics results in pronounced performance drops in $\bar{\mathcal{A}}$ (Average Accuracy across incremental tasks) for MOS and RAPF.

of access to static, pre-collected datasets and consequently struggle to generalize when data distributions evolve over time. In contrast, real-world environments are inherently dynamic: new concepts continuously emerge, while previously acquired knowledge may become outdated. To operate effectively under such non-stationary conditions, learning systems must continuously acquire new knowledge while retaining what has been learned before.

Continual learning addresses this challenge by enabling models to incrementally acquire, update, and accumulate knowledge from a sequence of tasks or data streams without assuming simultaneous access to all training data. A central difficulty in this setting is catastrophic forgetting, where adapting to new tasks overwrites previously learned knowledge, leading to severe performance degradation on earlier tasks. Traditional approaches mitigate this issue through regularization (Kirkpatrick et al., 2017; Rebuffi et al., 2017), rehearsal (Lopez-Paz & Ranzato, 2017), or representation learning (Cha et al., 2021), often in learning-from-scratch settings with randomly initialized parameters. Without strong prior representations, balancing plasticity and stability remains particularly challenging.

Recent advances in large-scale pre-trained models (PTMs) have substantially reshaped class-incremental learning (CIL). Models pre-trained on massive datasets provide rich and transferable representations, enabling strong performance even under severe data access constraints. Building

[1] Sichuan University, China [2] Engineering Research Center of Machine Learning and Industry Intelligence, Ministry of Education, China [3] MPU, Macao, China [4] UESTC, China [5] CQUPT, China [6] Key Laboratory of Cyberspace Big Data Intelligent Security, Ministry of Education, China [7] Ant Group, China. Correspondence to: Shudong Huang <huangsd@scu.edu.cn>.

*Proceedings of the 43rd International Conference on Machine Learning*, Seoul, South Korea. PMLR 306, 2026. Copyright 2026 by the author(s).

on such representations, many recent high-performing CIL methods (Zhang et al., 2023; Wang et al., 2023a; Sun et al., 2025; Wang et al., 2025)—particularly those based on ViT backbones (Dosovitskiy, 2021)—achieve results approaching static joint training under strictly incremental protocols.

Despite their architectural differences, a closer examination reveals a common design pattern among these methods: class-level feature statistics are incrementally estimated and used to synthesize representative features that support learning across tasks. This strategy is also prevalent in multimodal settings, including CLIP-based continual learning (Huang et al., 2024; Wen et al., 2025), where statistics-driven feature replay is used to adapt lightweight task-specific modules. While empirically effective, the specific role played by such replay mechanisms under strong pre-training remains insufficiently understood.

To better characterize this role, we conduct a pilot study on representative methods such as RAPF (Huang et al., 2024) and MOS (Sun et al., 2025). As shown in Figure 1, removing statistics-driven feature replay (wo Stat) leads to substantial performance degradation. Importantly, this comparison does not aim to contrast replay versus no replay in general, but to isolate the effect of statistics-derived synthetic features as a surrogate for historical data. The pronounced drop indicates that, under strong pre-training, such synthetic features closely approximate past feature distributions, and replaying these approximations accounts for a significant fraction of the recovered performance.

Motivated by this observation, we adopt a deliberately minimal perspective that treats class-level feature statistics as a standalone source of information under strong pre-training. Rather than viewing replay as an architectural necessity, we isolate the contribution of class-level feature statistics through two controlled reference points: a replay-based setting that uses statistics-derived synthetic features for lightweight adaptation, and a replay-free setting that performs class-incremental inference directly from class-level statistical estimators.

Concretely, we consider *SCLIP* as a minimal replay-based reference built on frozen CLIP representations, where a lightweight visual module is adapted using synthetic features sampled from incrementally estimated class-wise distributions. Despite its simplicity, SCLIP approaches static joint-training performance across multiple benchmarks, indicating that class-level statistics can account for a substantial fraction of the recovered performance under strong pre-training. At the other extreme, we consider *SViT*, a replay-free inference reference that performs class-incremental classification by operating directly on class-level statistical estimators instantiated from frozen features, including class prototypes and class-conditional density estimators. Together, these two references provide a controlled lens for ex-

amining how class-level statistics support class-incremental learning independently of architectural complexity.

Our contributions are twofold: (1) We empirically examine the role of class-level feature statistics in recent PTM-based class-incremental learning methods, showing that such statistics represent an important and often under-isolated component of their effectiveness under strong pre-training. This analysis offers a simple statistical perspective for interpreting a broad range of replay- and statistics-based approaches. (2) We instantiate this perspective through two deliberately minimal reference points—one replay-based and one replay-free—that isolate different ways class-level statistics are exploited in practice, ranging from statistics-driven adaptation with synthetic features to direct statistical inference without continual parameter updates.

## 2. Related Work

**Class-Incremental Learning (CIL):** aims to continuously adapt to evolving data streams by building a unified classifier over all seen categories. Classical CIL methods typically follow a learn-from-scratch paradigm and can be broadly categorized into several families. *Replay-based* methods store a subset of old data or reconstruct historical data distributions for rehearsal (Lopez-Paz & Ranzato, 2017). *Knowledge distillation-based* methods transfer knowledge from previous models to the current model and can be further divided into logit (Li & Hoiem, 2017), feature (Hou et al., 2019), and relational (Dong et al., 2021) distillation approaches. *Regularization-based* methods identify parameters critical to past tasks and impose constraints to prevent their drift (Kirkpatrick et al., 2017). Finally, *dynamic network-based* methods allocate additional modules or expand network components to accommodate new tasks, facilitating continual adaptation (Yan et al., 2021; Douillard et al., 2022). SCLIP samples features from class-level statistics and is most closely related to *generative replay*. Further discussion is provided in Appendix A.

**Pre-Trained Model-Based CIL:** leverages foundation models to initialize a continual learner, enabling efficient adaptation to non-stationary data streams. These pre-trained models, typically trained on large-scale datasets with sophisticated architectures, possess strong generalization capabilities that significantly reduce the learning burden in CIL (Wu et al., 2022). Integration strategies can be roughly divided into two categories: (1) full fine-tuning of the pre-trained model (Zhang et al., 2023; Zheng et al., 2023), or (2) freezing the backbone and introducing lightweight, trainable modules (Gao et al., 2023; Huang et al., 2025). Full fine-tuning is often computationally demanding and risks overwriting valuable pre-trained knowledge, leading to catastrophic forgetting. To address this, recent research has emphasized Parameter-Efficient Fine-Tuning strategies (He

et al., 2022a). Among these, *prompt-based* methods encode task-specific knowledge by injecting learnable tokens into the token sequence (Wang et al., 2022b;a; Smith et al., 2023; Wang et al., 2023b). These prompts can be introduced at different stages of the encoder—ranging from shallow to deep layers—and interact with the evolving representations to modulate feature extraction and adaptation across tasks. *Adapter-based* methods insert small neural modules into the pre-trained architecture, offering a favorable trade-off between parameter efficiency and continual learning performance (Zhou et al., 2024; 2025a;b;c; He et al., 2025). These strategies have been widely adopted in CIL to mitigate forgetting while maintaining adaptability.

In addition to these adaptation mechanisms, many PTM-based CIL methods also maintain class-level statistics and employ feature replay as an auxiliary means to support learning across tasks (Sun et al., 2025; Wang et al., 2025; Huang et al., 2024; Wen et al., 2025). While this practice has been widely adopted and shown to be effective, it is typically treated as an implementation detail for stabilizing training, and its role under strong pre-training has received limited analysis.

# 3. Preliminaries

We consider the *class-incremental learning* setting, where training data arrive sequentially as a series of tasks $\{\mathcal{D}^1, \ldots, \mathcal{D}^B\}$. Each task $\mathcal{D}^b = \{(\mathbf{x}_i, y_i)\}_{i=1}^{n_b}$ contains samples from a task-specific class set $Y_b$, with $Y_b \cap Y_{b'} = \varnothing$ for $b \neq b'$. The goal of CIL is to learn a single classifier that recognizes all classes observed so far, i.e., over the cumulative label space $\mathcal{Y}_b = \bigcup_{i=1}^{b} Y_i$. Formally, the objective is to learn a model $f : \mathcal{X} \to \mathcal{Y}_b$ that minimizes the expected classification risk over all data observed up to task $b$.

We adopt a pre-trained CLIP model as the backbone classifier. CLIP consists of a visual encoder $g_i(\cdot)$ and a text encoder $g_t(\cdot)$, which map image and text inputs into a shared $d$-dimensional embedding space. Given an input image $\mathbf{x}$, classification is performed by matching its visual embedding against text embeddings constructed from class names:

$$f_{y_i}(\mathbf{x}) = \frac{\exp\big(\cos\big(g_i(\mathbf{x}), g_t(\mathbf{t}_i)\big)/\tau\big)}{\sum_{j=1}^{|\mathcal{Y}_b|} \exp\big(\cos\big(g_i(\mathbf{x}), g_t(\mathbf{t}_j)\big)/\tau\big)}, \quad (1)$$

where $\cos(\cdot, \cdot)$ denotes cosine similarity, $\tau$ is a temperature parameter, and $\mathbf{t}_i$ is a prompt constructed from the class name (e.g., "a photo of a [CLASS]").

# 4. Statistical Modeling under Strong PTMs

This section does not propose a new continual learning algorithm. Instead, we conduct a controlled analysis of how class-wise feature statistics— which are widely but often implicitly exploited in PTM-based CIL— contribute to class-incremental performance under strong visual representations. By progressively examining replay-based adaptation, density-driven inference, and replay-free decision rules, we aim to clarify when simple statistical summaries are sufficient, and where their limitations arise.

## 4.1. Statistics as Supervision

We begin with **SCLIP**, a deliberately minimal class-incremental learning reference designed to explicitly expose the role of class-level statistical modeling under strong pre-trained representations. Rather than introducing new architectural components, SCLIP is constructed as a controlled reference point that isolates the contribution of class-wise statistics, allowing us to examine how much continual adaptation can be attributed to statistics-driven replay alone when the backbone representation is fixed.

Modern vision foundation models are known to induce compact and semantically structured feature spaces. Accordingly, prior work in PTM-based CIL often assumes that class-conditional representations are approximately unimodal and can be reasonably modeled by a single Gaussian (Zhang et al., 2023; Wang et al., 2023a). This assumption underlies many statistics-based replay mechanisms and implicit distributional approximations in prior work, making it a natural starting point for analysis.

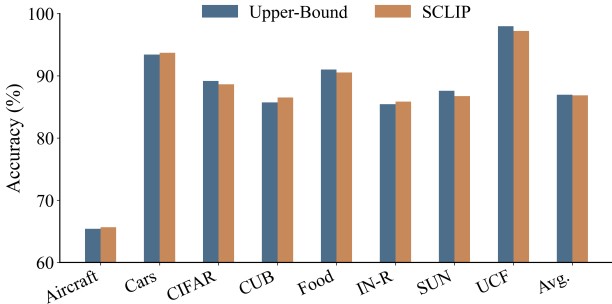

*Figure 2.* Average accuracy $\bar{\mathcal{A}}$ of SCLIP relative to the static joint-training upper bound across datasets.

As illustrated in Figure 3 (top), we consider a standard class-incremental setting in which data arrive sequentially across tasks. Images are processed by a frozen visual encoder $g_i(\cdot)$; after estimating class-level statistics, the extracted features are discarded. For each class $c$, we retain only empirical first- and second-order feature statistics, modeling class-conditional features as a unimodal Gaussian:

$$\mu_c = \frac{1}{n_c} \sum_{i=1}^{n_c} g_i(\mathbf{x}_i), \quad (2)$$

$$\Sigma_c = \frac{1}{n_c - 1} \sum_{i=1}^{n_c} \big(g_i(\mathbf{x}_i) - \mu_c\big)\big(g_i(\mathbf{x}_i) - \mu_c\big)^{\top}. \quad (3)$$

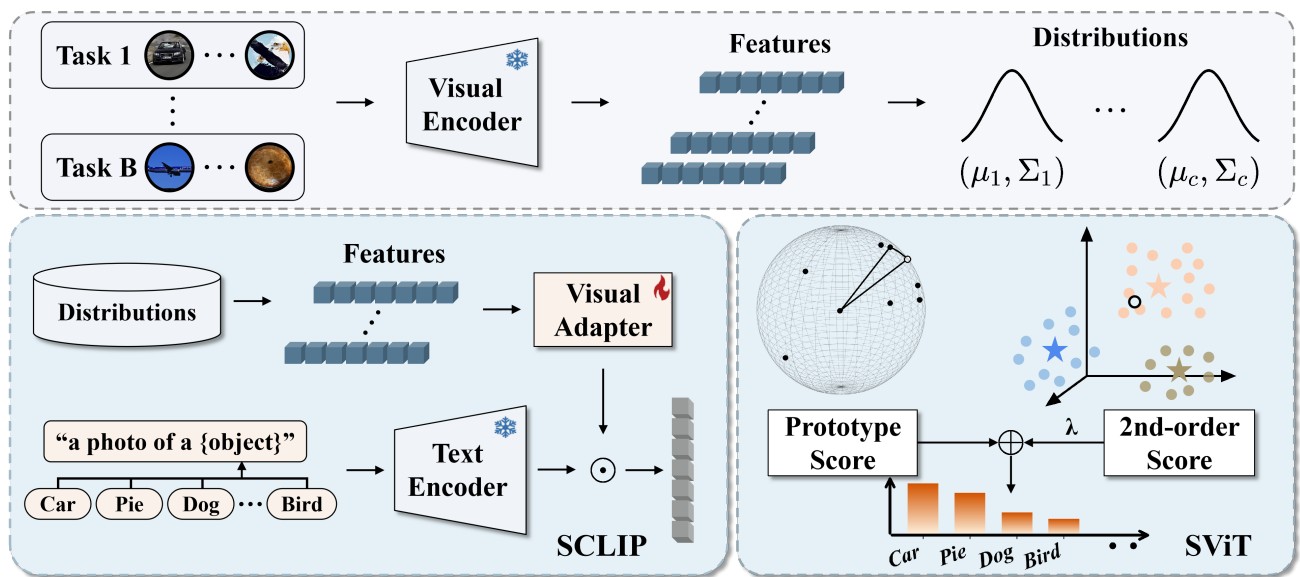

Figure 3. Statistical references under class-incremental learning with frozen representations. Top: The class-incremental setting, where data arrive sequentially across tasks. Images are encoded by a frozen visual backbone, and raw samples are discarded after each stage; only compact class-level information derived from features is retained. Bottom-left (SCLIP): A replay-based statistical reference that incrementally maintains class-wise feature distributions and samples synthetic features to support lightweight adapter training. Bottom-right (SViT): A replay-free statistical inference reference in which, for each class, a prototype and a class-conditional GMM are instantiated from that class's features and jointly used for inference via similarity- and density-based scores.

A small diagonal regularization $\Sigma_c \leftarrow \Sigma_c + \alpha I$ with $\alpha = 10^{-4}$ is applied for numerical stability.

Based on these statistics, SCLIP adapts to new classes using a lightweight linear visual adapter (Figure 3, bottom-left). At each incremental stage, synthetic features sampled from all stored class-wise distributions are used to train the adapter, serving as a compact substitute for historical data. This update–replay cycle is repeated throughout training, with statistics updated using newly observed samples and reused for subsequent adaptation.

Figure 2 compares SCLIP against the static joint-training upper bound across multiple benchmarks. Despite its simplicity, SCLIP consistently approaches this upper bound, indicating that under strong pre-training, class-level feature statistics preserve a substantial fraction of the information required for effective class-incremental classification. Additional implementation details and variants are provided in Appendix B, with further analysis under different pre-training paradigms in Appendix D.

### 4.2. Limits of Density-Based Inference

While effective, SCLIP still relies on feature generation and replay. In more restrictive continual learning settings, even feature-space replay may be disallowed. This raises a natural question: *can class-wise statistics be exploited directly for inference, without replay or additional parameter learning?*

A straightforward replay-free alternative is to perform inference directly in feature space based on estimated class-conditional densities. Given class-wise statistics, the Mahalanobis distance measures how well a sample conforms to a class distribution:

$$d^{\text{Maha}}(\mathbf{x}, c) = \big(g_i(\mathbf{x}) - \mu_c\big)^\top \Sigma_c^{-1} \big(g_i(\mathbf{x}) - \mu_c\big). \quad (4)$$

By reweighting deviations along different feature directions, this distance accounts for anisotropic class geometry. All covariance matrices used for Mahalanobis inference are inherited directly from the statistics estimated in Section 4.1, including the same diagonal regularization.

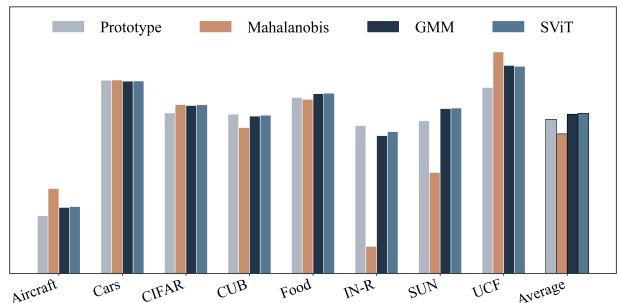

Figure 4. Comparison of different class-conditional distribution estimators applied to the same frozen feature space, differing only in how class-level statistics are estimated and used for inference.

However, despite its simplicity and theoretical appeal, Mahalanobis-based inference exhibits severe performance

degradation on several benchmarks, particularly ImageNet-R and SUN, as shown in Figure 4.

A natural follow-up question is whether this failure mainly stems from the unimodal Gaussian assumption underlying Mahalanobis distance, which may be overly restrictive for complex class-conditional feature distributions. To examine this possibility, we replace the single Gaussian assumption with a more expressive yet still lightweight Gaussian Mixture Model (GMM):

$$p(\mathbf{x} \mid y = c) = \sum_{k=1}^{K_c} \pi_{c,k} \mathcal{N}\big(\mathbf{x} \mid \mu_{c,k}, \Sigma_{c,k}\big). \qquad (5)$$

The number of mixture components $K_c \in \{1, \ldots, 5\}$ is selected using the Bayesian Information Criterion (BIC) (Schwarz, 1978), with the same diagonal regularization used in Section 4.1 for numerical stability. Throughout this work, $\Sigma_{c,k}$ is restricted to be diagonal. Empirically, under strong pre-trained representations, allowing full covariance matrices consistently causes model selection to favor a single mixture component, leading the GMM to degenerate into a unimodal Gaussian. Imposing a diagonal constraint prevents this collapse, while retaining sufficient flexibility to capture multi-modal structure.

| Dataset | G-LL ↑ | GMM-LL ↑ | Skew ↓ | Kurt ≈ 3 | Sil. ↑ |
|---------|--------|----------|--------|----------|--------|
| IN-R    | 365.0  | 1039.4   | 0.24   | 3.01     | 0.096  |
| UCF     | 1357.9 | 1214.1   | 0.29   | 2.96     | 0.208  |

*Table 1.* Statistical diagnostics of class-wise feature distributions. For each dataset, five classes are randomly sampled and evaluated over multiple trials, with the average results reported. See Appendix E for details of the reported metrics.

As shown in Table 1, moving from a unimodal Gaussian to a GMM substantially improves held-out log-likelihood on ImageNet-R, revealing pronounced multi-modal structure in the class-conditional feature distributions. This improvement also translates into consistent gains over Mahalanobis inference across several benchmarks (Figure 4), particularly on ImageNet-R and SUN. Nevertheless, improved density expressiveness alone does not yield a consistently effective inference strategy, suggesting that the failure is not solely caused by inaccurate density fitting. Surprisingly, even a much simpler prototype-based rule, which entirely ignores intra-class covariance structure, remains competitive or superior on several benchmarks including CUB and Cars.

To further understand this discrepancy, we analyze the geometry of the frozen feature space. Although datasets such as UCF and ImageNet-R exhibit similar skewness and kurtosis values close to the Gaussian reference, their inference behavior differs markedly. In particular, ImageNet-R attains substantially lower silhouette scores, indicating weak inter-class separability despite approximately Gaussian marginals.

This observation is consistent with the feature visualizations in Figure 5, where representations from different ImageNet-R classes overlap heavily, while UCF exhibits clearer semantic separation.

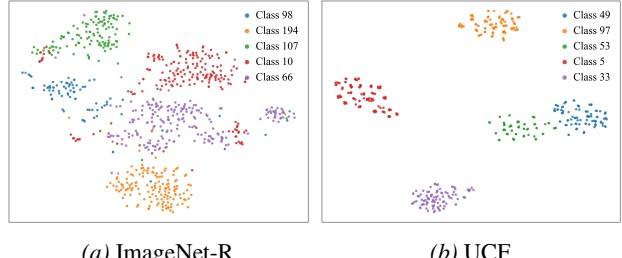

*(a)* ImageNet-R           *(b)* UCF

*Figure 5.* t-SNE visualization illustrating inter-class overlap under different datasets.

Taken together, these results suggest that the core limitation does not primarily arise from insufficient density expressiveness, but from the density-based inference paradigm itself. Even when estimated densities capture rich local structure, likelihood-based decisions remain sensitive to class overlap and may produce unstable or overly confident predictions. By contrast, prototype-based similarity provides a complementary global semantic bias anchored in the organization of the pre-trained feature space. However, prototype-based inference also discards fine-grained local geometry by collapsing each class into a single centroid, making it less effective for classes with complex intra-class variability, as reflected by its clear performance degradation on CIFAR and UCF in Figure 4.

### 4.3. Inference without Replay

Motivated by the complementary strengths and failure modes of prototype-based and density-based inference, we introduce **SViT**, a deliberately simple and fully replay-free statistical inference reference.

Unlike SCLIP, which relies on replay-based adaptation, SViT completely removes replay and continual parameter updates, focusing purely on inference-time statistical composition. At each incremental stage, for every newly observed class, SViT instantiates two complementary class-specific inference primitives: (i) a class prototype defined as the mean feature vector, used for cosine-similarity inference, and (ii) a class-conditional GMM, estimated from the feature vectors of that class, as described in Section 4.2. Once instantiated, these per-class prototypes and density estimates are retained and jointly used for inference over all seen classes (Figure 3, bottom-right).

Given an input image $\mathbf{x}$, SViT computes a prototype-based

| Method | Aircraft | Cars | CIFAR | CUB | Food | IN-R | SUN | UCF | Average |
|---|---|---|---|---|---|---|---|---|---|
| Upper-Bound | 65.40 | 93.41 | 89.16 | 85.71 | 91.01 | 85.45 | 87.58 | 97.97 | 86.96 |
| RAPF | 59.28 | 92.58 | 87.47 | 84.24 | 89.48 | 85.11 | 85.95 | 95.05 | 84.49 |
| PROOF | 64.61 | 92.70 | 86.77 | 84.87 | 90.04 | 84.19 | 83.75 | 94.34 | 84.16 |
| SCLIP$^-$ | 61.37 | 93.26 | 86.61 | 84.72 | 88.94 | 84.14 | 84.52 | 95.06 | 84.30 |
| SCLIP | **65.67** | **93.69** | **88.64** | **86.53** | **90.54** | **85.85** | **86.74** | **97.22** | **86.86** |

*Table 2.* Average accuracy $\bar{\mathcal{A}}$ of SCLIP and competing methods. The best results excluding the Upper-Bound are highlighted in bold.

similarity score:

$$s_c^{\text{proto}}(\mathbf{x}) = \frac{g_i(\mathbf{x})^\top \mu_c}{\|g_i(\mathbf{x})\|_2 \|\mu_c\|_2}. \quad (6)$$

In parallel, a complementary statistical cue is obtained via class-conditional density estimation, using the log-likelihood under the class-specific GMM:

$$s_c^{\text{2nd}}(\mathbf{x}) = \log p(g_i(\mathbf{x}) \mid y = c). \quad (7)$$

For numerical comparability, log-likelihood scores are normalized across classes on a per-sample basis using a softmax transformation. Final predictions are obtained via simple additive fusion:

$$\hat{y} = \arg\max_{c \in \mathcal{Y}_b} \left( s_c^{\text{proto}}(\mathbf{x}) + \lambda s_c^{\text{2nd}}(\mathbf{x}) \right), \quad (8)$$

with the fusion weight fixed to $\lambda = 0.1$ in all experiments; performance is empirically insensitive to moderate variations of this value. The formal definition of SViT is provided in Algorithm 1.

---

**Algorithm 1** SViT for CIL

---

**Input:** Incremental datasets $\{\mathcal{D}^1, \mathcal{D}^2, \ldots, \mathcal{D}^B\}$; frozen CLIP visual encoder $g_i(\cdot)$
**Output:** The complementary classifier
**for** $b = 1$ **to** $B$ **do**
    Get the incremental training set $\mathcal{D}^b$
    Extract visual features $g_i(\mathbf{x})$ for all samples in $\mathcal{D}^b$
    **for** each $c \in Y_b$ **do**
        Compute per-class prototype via Equation (2)
        Fit GMM to per-class features via Equation (5)
    **end for**
    Construct the complementary classifier as Equation (8)
**end for**

---

As shown in Figure 4, although SViT is intentionally simple and entirely replay-free, it achieves competitive performance across diverse benchmarks by combining prototype-based semantic alignment with density-based statistical estimation. In particular, these results demonstrate that competitive class-incremental performance can be achieved without

continual parameter updates, using only class-level statistical estimators instantiated from frozen representations.

Overall, SViT is not intended as a sophisticated inference framework, but rather as a minimal statistical reference for understanding how much continual learning capability is already embedded within strong pre-trained feature spaces.

## 5. Experiments

### 5.1. Implementation Details

**Datasets:** We evaluate our methods on eight benchmark datasets that exhibit clear domain gaps from CLIP's pre-training data, including FGVCAircraft (Maji et al., 2013), StanfordCars (Krause et al., 2013), CIFAR100 (Krizhevsky et al., 2009), CUB200 (Wah et al., 2011), Food101 (Bossard et al., 2014), ImageNet-R (Hendrycks et al., 2021), SUN397 (Xiao et al., 2010), and UCF101 (Soomro et al., 2012). Following (Zhou et al., 2025b), we utilize the sampled 100 classes from CIFAR100, Aircraft, Cars, Food, and UCF101, 200 classes from CUB200 and ImageNet-R, and 300 classes from SUN397 to facilitate incremental splits. ImageNet-R is sometimes simply referred to as IN-R.

**Incremental Protocol:** We adopt the `Base-m Inc-n` protocol, where $m$ denotes the number of base classes and $n$ the number of classes introduced per incremental task. When $m = 0$, all tasks contain an equal number of classes. Unless otherwise specified, we adopt the `Base-0` protocol with classes evenly split into 10 tasks, and fix the class order using random seed 1993 following (Rebuffi et al., 2017). Results in Table 3 are reported as the mean and standard deviation over three runs with seeds 1992, 1993, and 1994; all other results use the default seed 1993.

**Compared Methods:** We compare our method with a diverse set of continual learning baselines, including L2P, DualPrompt, CODA-Prompt, APER-Adapter, EASE, RAPF, PROOF, and MGCLIP. Detailed descriptions of the compared methods and additional experimental results are provided in Appendix F, Appendix G and Appendix I, respectively. For vision-only methods, the CLIP visual tower is used as initialization. All pre-trained weights come from the LAION-400M pre-trained CLIP for fairness.

| Method | Aircraft | | | | Cars | | | |
| --- | --- | --- | --- | --- | --- | --- | --- | --- |
| | B-0Inc-10 | | B-50Inc-10 | | B-0Inc-10 | | B-50Inc-10 | |
| | $\bar{\mathcal{A}}$ | $\mathcal{A}_B$ | $\bar{\mathcal{A}}$ | $\mathcal{A}_B$ | $\bar{\mathcal{A}}$ | $\mathcal{A}_B$ | $\bar{\mathcal{A}}$ | $\mathcal{A}_B$ |
| L2P | $46.92 \pm 2.01$ | $30.36 \pm 1.50$ | $47.97 \pm 1.44$ | $33.29 \pm 1.43$ | $72.21 \pm 3.66$ | $58.28 \pm 2.60$ | $78.91 \pm 3.99$ | $68.75 \pm 3.58$ |
| DualPrompt | $44.01 \pm 3.76$ | $25.27 \pm 2.68$ | $44.91 \pm 2.46$ | $31.76 \pm 1.88$ | $68.96 \pm 1.74$ | $53.14 \pm 1.26$ | $73.48 \pm 2.19$ | $61.54 \pm 2.24$ |
| CODA-Prompt | $47.75 \pm 5.03$ | $29.32 \pm 2.59$ | $48.39 \pm 2.07$ | $31.97 \pm 2.15$ | $84.05 \pm 2.00$ | $74.12 \pm 0.98$ | $78.13 \pm 1.36$ | $66.32 \pm 0.98$ |
| APER-Adapter | $61.64 \pm 1.54$ | $49.28 \pm 0.05$ | $\mathbf{57.66 \pm 0.83}$ | $\mathbf{52.58 \pm 0.35}$ | $91.66 \pm 0.83$ | $86.73 \pm 0.02$ | $\mathbf{89.82 \pm 0.86}$ | $\mathbf{87.58 \pm 0.10}$ |
| EASE | $60.93 \pm 1.42$ | $48.01 \pm 0.51$ | $40.75 \pm 1.87$ | $36.43 \pm 1.28$ | $91.56 \pm 0.96$ | $86.56 \pm 0.33$ | $89.31 \pm 0.65$ | $86.59 \pm 0.22$ |
| MGCLIP | $39.60 \pm 2.88$ | $29.09 \pm 0.93$ | $36.34 \pm 1.21$ | $30.35 \pm 0.98$ | $86.82 \pm 1.43$ | $79.42 \pm 0.87$ | $85.36 \pm 0.95$ | $81.04 \pm 1.13$ |
| SViT | $\mathbf{62.96 \pm 1.43}$ | $\mathbf{50.83 \pm 0.47}$ | $55.54 \pm 0.89$ | $50.83 \pm 0.47$ | $\mathbf{91.73 \pm 1.27}$ | $\mathbf{87.22 \pm 0.34}$ | $89.45 \pm 1.24$ | $87.22 \pm 0.34$ |

| Method | CUB | | | | Food | | | |
| --- | --- | --- | --- | --- | --- | --- | --- | --- |
| | B-0Inc-20 | | B-100Inc-20 | | B-0Inc-10 | | B-50Inc-10 | |
| | $\bar{\mathcal{A}}$ | $\mathcal{A}_B$ | $\bar{\mathcal{A}}$ | $\mathcal{A}_B$ | $\bar{\mathcal{A}}$ | $\mathcal{A}_B$ | $\bar{\mathcal{A}}$ | $\mathcal{A}_B$ |
| L2P | $70.56 \pm 1.11$ | $58.73 \pm 1.82$ | $56.26 \pm 2.64$ | $40.23 \pm 2.58$ | $76.29 \pm 1.75$ | $65.30 \pm 3.29$ | $71.00 \pm 0.84$ | $59.33 \pm 2.33$ |
| DualPrompt | $63.82 \pm 1.38$ | $48.91 \pm 2.09$ | $53.31 \pm 1.50$ | $37.71 \pm 1.56$ | $79.11 \pm 1.53$ | $69.38 \pm 0.94$ | $70.24 \pm 0.99$ | $58.88 \pm 2.66$ |
| CODA-Prompt | $76.17 \pm 1.24$ | $65.42 \pm 0.86$ | $71.01 \pm 1.25$ | $58.18 \pm 2.01$ | $85.35 \pm 1.26$ | $77.47 \pm 0.54$ | $83.30 \pm 0.98$ | $76.91 \pm 1.92$ |
| APER-Adapter | $82.87 \pm 0.46$ | $76.40 \pm 0.05$ | $\mathbf{81.56 \pm 0.19}$ | $\mathbf{79.20 \pm 0.02}$ | $86.97 \pm 0.82$ | $81.53 \pm 0.21$ | $85.26 \pm 0.69$ | $82.90 \pm 0.08$ |
| EASE | $82.46 \pm 0.53$ | $75.90 \pm 0.27$ | $54.46 \pm 1.18$ | $50.35 \pm 1.20$ | $86.71 \pm 0.76$ | $80.97 \pm 0.22$ | $80.50 \pm 1.01$ | $75.82 \pm 0.62$ |
| MGCLIP | $75.07 \pm 1.33$ | $65.40 \pm 0.69$ | $72.42 \pm 1.18$ | $67.02 \pm 0.92$ | $88.90 \pm 0.66$ | $\mathbf{83.82 \pm 0.20}$ | $\mathbf{86.85 \pm 0.49}$ | $\mathbf{84.16 \pm 0.33}$ |
| SViT | $\mathbf{83.33 \pm 0.30}$ | $\mathbf{77.14 \pm 0.15}$ | $79.63 \pm 0.35$ | $77.14 \pm 0.15$ | $\mathbf{88.91 \pm 0.24}$ | $83.47 \pm 0.08$ | $86.15 \pm 0.31$ | $83.47 \pm 0.08$ |

| Method | SUN | | | | UCF | | | |
| --- | --- | --- | --- | --- | --- | --- | --- | --- |
| | B-0Inc-30 | | B-150Inc-30 | | B-0Inc-10 | | B-50Inc-10 | |
| | $\bar{\mathcal{A}}$ | $\mathcal{A}_B$ | $\bar{\mathcal{A}}$ | $\mathcal{A}_B$ | $\bar{\mathcal{A}}$ | $\mathcal{A}_B$ | $\bar{\mathcal{A}}$ | $\mathcal{A}_B$ |
| L2P | $70.85 \pm 1.85$ | $57.03 \pm 3.77$ | $62.75 \pm 0.74$ | $47.18 \pm 1.23$ | $82.94 \pm 1.56$ | $72.85 \pm 2.93$ | $83.06 \pm 5.93$ | $73.69 \pm 9.03$ |
| DualPrompt | $75.19 \pm 0.74$ | $63.12 \pm 1.12$ | $62.84 \pm 0.88$ | $47.72 \pm 0.62$ | $84.26 \pm 2.18$ | $73.83 \pm 1.98$ | $73.30 \pm 2.93$ | $57.33 \pm 4.75$ |
| CODA-Prompt | $84.68 \pm 0.56$ | $76.30 \pm 0.57$ | $76.87 \pm 0.50$ | $68.04 \pm 1.10$ | $90.02 \pm 1.23$ | $83.60 \pm 0.65$ | $87.27 \pm 0.27$ | $78.09 \pm 1.37$ |
| APER-Adapter | $83.51 \pm 0.74$ | $76.45 \pm 0.11$ | $80.85 \pm 0.73$ | $77.60 \pm 0.14$ | $89.50 \pm 0.83$ | $85.86 \pm 0.06$ | $88.77 \pm 0.58$ | $87.41 \pm 0.21$ |
| EASE | $84.18 \pm 0.69$ | $77.35 \pm 0.27$ | $79.27 \pm 0.62$ | $75.10 \pm 0.46$ | $89.61 \pm 0.72$ | $85.92 \pm 0.23$ | $80.90 \pm 1.84$ | $77.73 \pm 1.66$ |
| MGCLIP | $84.93 \pm 0.64$ | $77.64 \pm 0.20$ | $\mathbf{82.90 \pm 0.44}$ | $78.75 \pm 0.50$ | $81.52 \pm 1.17$ | $73.77 \pm 0.64$ | $83.37 \pm 1.56$ | $77.91 \pm 1.01$ |
| SViT | $\mathbf{86.12 \pm 0.71}$ | $\mathbf{79.71 \pm 0.18}$ | $82.84 \pm 0.60$ | $79.71 \pm 0.18$ | $\mathbf{95.03 \pm 0.70}$ | $\mathbf{92.34 \pm 0.33}$ | $\mathbf{93.55 \pm 0.59}$ | $\mathbf{92.34 \pm 0.33}$ |

*Table 3.* Average accuracy $\bar{\mathcal{A}}$ and final accuracy $\mathcal{A}_B$ of SViT and competing methods. The best results are highlighted in bold.

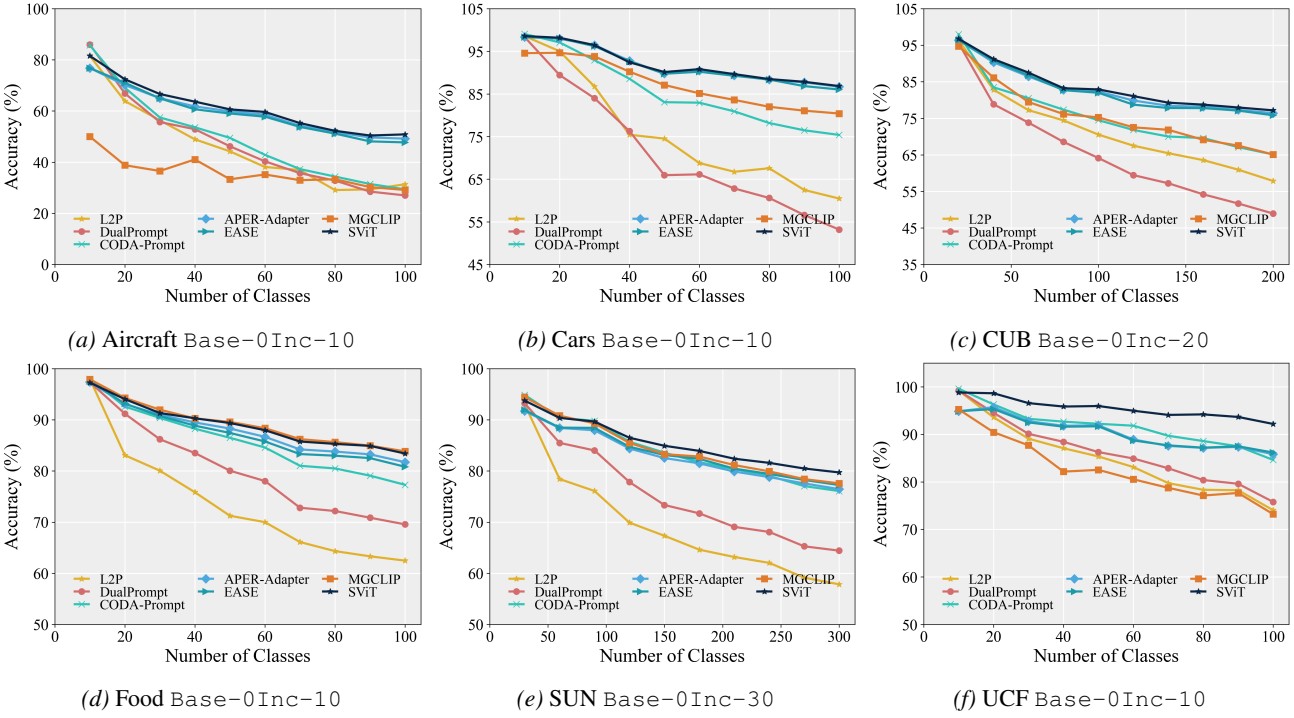

*(a)* Aircraft `Base-0Inc-10`

*(b)* Cars `Base-0Inc-10`

*(c)* CUB `Base-0Inc-20`

*(d)* Food `Base-0Inc-10`

*(e)* SUN `Base-0Inc-30`

*(f)* UCF `Base-0Inc-10`

*Figure 6.* Incremental performance of different methods under `Base-0` settings.

**Training Details:** All experiments are conducted in Py-Torch on a single NVIDIA RTX 3090 GPU. For SCLIP and its variants, trainable parameters are optimized using SGD for 10 epochs with batch size 256, initial learning rate 0.05, and cosine annealing.

**Evaluation Metrics:** Following (Zhou et al., 2025c), we denote the test accuracy after learning the $b$-th task as $\mathcal{A}_b$. We report two standard metrics: the final accuracy $\mathcal{A}_B$ after learning all tasks, and the average accuracy $\bar{\mathcal{A}} = \frac{1}{B}\sum_{b=1}^{B}\mathcal{A}_b$ across all incremental tasks.

### 5.2. Benchmark Comparison

We first compare SCLIP with two representative CLIP-based methods, RAPF and PROOF, on eight benchmark datasets under the `Base-0` protocol, as summarized in Table 2. To contextualize the results, we additionally report SCLIP$^-$, a lightweight variant in which the number of synthesized features per class is reduced to 16.

RAPF and PROOF both rely on replay-based strategies. Notably, even SCLIP$^-$, which uses a highly limited number of synthetic features, exhibits performance comparable to these methods across most benchmarks. This observation suggests that a large portion of the effectiveness of replay-based PTM-CIL methods can be attributed to coarse class-level feature statistics, rather than to the scale or complexity of replayed features. Together with the pilot results in Figure 1, this supports the view that approximating and reusing class-level feature distributions constitutes a central operational mechanism in many recent approaches.

While effective, replay-based adaptation still requires feature generation and additional training, which may be undesirable in strictly constrained continual learning settings. We therefore further examine SViT as a replay-free statistical inference reference, and compare it with several recent methods in Table 3 and Figure 6.

As shown in Table 3, SViT achieves competitive performance on UCF and remains comparable to existing methods on the remaining benchmarks. The results indicate that combining prototype-based similarity with class-conditional density estimates provides a stable and effective replay-free inference rule under strong pre-trained representations, without requiring continual parameter updates. In contrast, prompt-based methods such as L2P and DualPrompt are sensitive to retrieval quality and tend to degrade under class-incremental evaluation, consistent with prior observations (Gao et al., 2023). CODA-Prompt partially mitigates this issue through attention-based weighting. Among the compared baselines, MGCLIP achieves strong results on specific datasets such as Food, while exhibiting larger variability across benchmarks. Overall, these results highlight the trade-offs between architectural mechanisms and purely statistics-driven inference under strong pre-training, suggesting that competitive class-incremental performance can be achieved through substantially simpler decision rules when representations are sufficiently structured.

**Parameter Sensitivity.** We examine the sensitivity of SViT to the fusion weight $\lambda$ balancing prototype similarity and GMM-based scores. Across all evaluated datasets, performance remains stable under moderate variations of $\lambda$, with only negligible differences observed. A detailed sensitivity analysis is provided in Appendix H.

**Memory Cost.** We compare the memory footprint of statistics-based continual learning with exemplar replay by expressing all storage costs in terms of an equivalent number of raw images. For CLIP ViT-B/16 with a 512-dimensional representation, storing per-class first- and second-order statistics in SCLIP requires approximately 1.05 MB per class. On SUN with 300 classes, this amounts to about 315 MB, corresponding to roughly 2,100 raw images under standard CLIP preprocessing.

For comparison, a commonly used fixed-size replay buffer of 2,000 images requires a similar memory budget (approximately 294 MB). In contrast, per-class replay with 20 exemplars per class scales linearly with the number of classes, resulting in 6,000 images (about 882 MB) on SUN. This highlights the less favorable scaling behavior of per-class replay on datasets with many classes.

For SViT, memory usage is further reduced by modeling each class with a small diagonal-covariance GMM. Each class stores $K_c(2d+1)$ parameters, with $K_c \leq 5$ selected by BIC. In practice, many datasets are approximately unimodal, and the fitted GMMs often reduce to a single component, yielding a storage cost of roughly $c \times 2d$ parameters—well below that of a single $512 \times 512$ projection layer.

## 6. Conclusion

In this work, we revisit class-incremental learning from a statistical perspective under large-scale pre-trained representations. Through two deliberately minimal references, SCLIP and SViT, we show that much of the performance achieved by recent PTM-based continual learning methods can be recovered using simple class-level feature distributions alone. These findings suggest that the effectiveness of modern PTM-based class-incremental learning may depend less on increasingly sophisticated adaptation mechanisms and more on the structured feature spaces induced by strong pre-training. Overall, our results highlight the value of understanding how representation geometry and class-level statistics jointly support stable continual adaptation.

**Limitations and Future Work.** Our analysis assumes that strong pre-trained representations induce feature spaces

where class-conditional distributions can be reasonably approximated by simple Gaussian statistics. Future work will explore more expressive distribution estimation strategies to better exploit class-level statistical information beyond Gaussian assumptions.

## Acknowledgements

This work was partially supported by the National Science Foundation of China under Grant 62376175, 22494712 and U2333211, the National Science Foundation of Sichuan Province under Grant 2025ZNSFSC0480, the 111 Project under Grant B21044, and the Science Fund for Creative Research Groups of Sichuan Province Natural Science Foundation under Grant 2024NSFTD0035.

## Impact Statement

This paper presents work whose goal is to advance the field of Machine Learning. There are many potential societal consequences of our work, none which we feel must be specifically highlighted here.

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

## A. Generative Replay

Numerous prior studies have investigated *generative replay* as a strategy to mitigate catastrophic forgetting in class-incremental learning. Traditional generative replay-based CIL frameworks typically maintain two components: a generative model that synthesizes samples from previously learned classes, and a discriminative model trained on both real and generated data. Early approaches mainly relied on GANs to generate past instances (Shin et al., 2017), while subsequent works explored more advanced generative architectures, including conditional GANs (Ostapenko et al., 2019; Xiang et al., 2019), variational autoencoders (Jiang et al., 2021), and, more recently, diffusion models (Jodelet et al., 2023; Gao & Liu, 2023).

Despite steady progress, generative replay has been shown to scale poorly to complex and large-scale visual data (Wang et al., 2021; Van de Ven et al., 2020). Training high-fidelity generators remains challenging, and errors in generation quality can accumulate across incremental stages. To alleviate these issues, several works propose to replay features or intermediate representations instead of raw images, reducing the burden on generative modeling (Liu et al., 2020). However, in learning-from-scratch settings, feature representations often change substantially over time, making it difficult to jointly preserve old features and adapt to new classes without interference. Moreover, when generative models themselves are updated sequentially, they are also susceptible to catastrophic forgetting, which necessitates additional mechanisms to stabilize both the generator and the classifier.

The emergence of large pre-trained vision models significantly alters this landscape. Strong pre-training yields robust and transferable representations that remain relatively stable in continual learning, alleviating the central challenge of feature replay (Wang et al., 2024). In this regime, recent CIL methods increasingly move away from explicit generative models and instead characterize class-conditional feature distributions using simple parametric statistics (Zhang et al., 2023; Huang et al., 2024; Goswami et al., 2024; Sun et al., 2025; Wang et al., 2025; Chen et al., 2025). By modeling and accumulating these statistics, such methods can effectively simulate feature replay without additional optimization. This shift is motivated by empirical observations that modern vision encoders induce well-structured and compact feature distributions, for which Gaussian-based approximations are often sufficient.

In contrast to classical generative replay, this *statistical feature replay* paradigm eliminates the need to train and maintain a separate generator, avoids the risk of generator forgetting, and enables inference directly in representation space. SCLIP is situated within this line of research. Rather than proposing a new generative mechanism, we provide a focused empirical analysis of feature replay based on estimated class-wise distributions under strong pre-training, and examine when such statistical replay is sufficient and when its underlying assumptions begin to break down.

## B. SCLIP and Variants

### B.1. Variants Definition

This section provides an overview of *SCLIP* and several closely related variants considered in our study. All statistics-based variants follow the same incremental procedure as SCLIP, as illustrated in Figure 3 (top and bottom-left), including frozen feature extraction, class-wise statistics accumulation, and statistics-driven synthetic feature generation for adapting a lightweight visual adapter. The formal definition of this procedure is given in Algorithm 2. The only exception is the *Upper-Bound*, which corresponds to static joint training with unrestricted access to all task data and therefore does not operate under class-incremental constraints. It serves as an oracle reference rather than a statistics-driven variant.

We consider the following variants. *k-shot* restricts statistics estimation to $k$ randomly selected samples per class. *Covariance* denotes the default SCLIP setting, where full covariance matrices are maintained, while *Variance* replaces them with diagonal covariance estimates. *Prototype* discards second-order statistics and performs inference using only class means via cosine similarity. *Imbalance* simulates biased replay, where 256 synthetic features are generated for newly introduced classes but only 16 features are generated for previously observed classes. Unless otherwise specified, 256 synthetic features are generated per class.

---

**Algorithm 2** Feature Replay for SCLIP

---

**Input:** Incremental datasets $\{\mathcal{D}^1, \mathcal{D}^2, \ldots, \mathcal{D}^B\}$; frozen CLIP visual encoder $g_i(\cdot)$; linear visual adapter $\mathcal{A}$
**Output:** Trained adapter $\mathcal{A}$ after $B$ tasks
Initialize an empty set of class-wise statistics $S$
**for** $b = 1$ **to** $B$ **do**
    Get the incremental training set $\mathcal{D}^b$
    Extract visual features $g_i(\mathbf{x})$ for all samples in $\mathcal{D}^b$
    **for** each $c \in Y_b$ **do**
        Compute per-class statistics via Equation (2) and (3)
        Update stored statistics: $S \leftarrow S \cup \{(\mu_c, \Sigma_c)\}_{c \in Y_b}$
    **end for**
    Construct an empty training set $\hat{\mathcal{D}}$
    **for** each $c \in \mathcal{Y}_b$ **do**
        Gaussian sampling via $(\tilde{\mathbf{x}}, c) \sim \mathcal{N}(\mu_c, \Sigma_c)$
        Update training set $\hat{\mathcal{D}}$: $\hat{\mathcal{D}} \leftarrow \hat{\mathcal{D}} \cup (\tilde{\mathbf{x}}, c)$
    **end for**
    Train the adapter $\mathcal{A}$ using the constructed $\hat{\mathcal{D}}$
**end for**

---

The adapter is implemented as a single $512 \times 512$ linear projection appended to the frozen CLIP visual encoder. It is trained in a CLIP-style manner, where adapted visual features are matched against fixed class text embeddings via cosine similarity, followed by a softmax normalization and a standard cross-entropy loss to encourage correct image–text alignment. At inference time, an input image is encoded by the frozen CLIP visual encoder, passed through the adapter, and classified by computing cosine similarities with fixed class text embeddings and selecting the maximum after softmax normalization. The overall inference procedure is summarized in Figure 7.

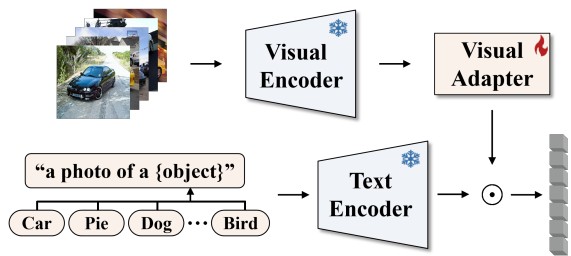

*Figure 7.* Inference pipeline of SCLIP. An input image is encoded by the frozen CLIP visual encoder, adapted by a lightweight linear adapter, and classified by computing cosine similarities with fixed class text embeddings.

### B.2. Discussion

Figure 8 reports the average $\bar{\mathcal{A}}$ results across eight datasets and analyzes how different forms of class-wise statistical modeling affect performance. As shown in the figure, replacing full covariance with diagonal covariance leads to a noticeable performance drop, highlighting the importance of modeling inter-dimensional correlations. A prototype-only variant remains competitive but consistently underperforms full covariance modeling, indicating that first-order statistics alone are insufficient to capture intra-class geometry. The figure further examines the effect of the number of real samples used to estimate class-wise statistics ($k$-shot), showing that improved statistical estimation consistently benefits downstream inference, while a moderate number of samples is sufficient to capture most of the performance gain. We list the detailed performance under different few-shot regimes in Table 4. Overall, using 32 samples per class is already sufficient to achieve strong performance across most datasets. For more challenging fine-grained or distribution-shifted benchmarks, such as Aircraft and ImageNet-R, additional samples further improve the quality of statistical estimation and lead to consistent performance gains.

Table 5 provides complementary evidence. We introduce SCLIP$^-$, a lightweight variant that synthesizes only 16 features per class, enabling a controlled study of how the number of generated features influences the effectiveness of statistical replay. A more detailed analysis of this factor is

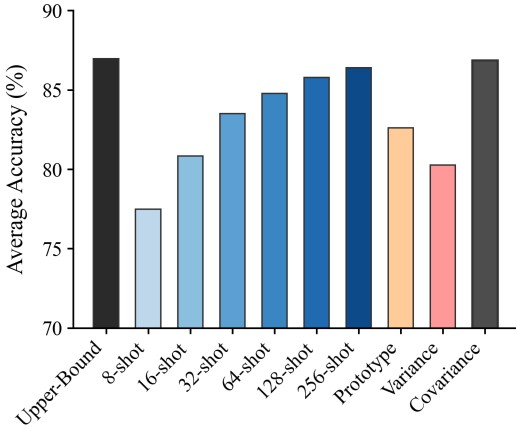

*Figure 8.* Effect of class-level statistical modeling and synthetic feature generation on class-incremental learning performance. Results are averaged over eight benchmarks.

provided in Appendix C. Despite its substantially reduced replay budget, SCLIP$^-$ maintains competitive performance across datasets. In contrast, *Imbalance* adopts an asymmetric replay strategy, where significantly more features are generated for new classes than for previously learned ones. Although both settings use the same number of samples for old classes, the excessive emphasis on new classes leads to inferior overall performance. This comparison suggests that maintaining a balanced replay across classes is crucial for effective continual learning.

## C. Effect of the Feature Number

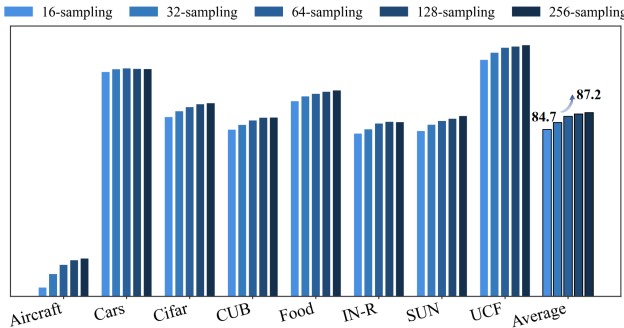

*Figure 9.* Effect of the number of generated features per class in SCLIP$^-$.

We further analyze the effect of the number of generated features per class in Figure 9. Under the default covariance setting, sampling only a small number of features per class already yields strong performance. As the number of generated samples increases, accuracy improves steadily and eventually saturates, indicating that a modest amount of feature-level replay is sufficient to recover most of the underlying class-conditional distribution.

| Method | Aircraft | Cars | CIFAR | CUB | Food | IN-R | SUN | UCF | Average |
|---|---|---|---|---|---|---|---|---|---|
| 8-shot | 49.68 | 90.19 | 79.04 | 78.36 | 82.91 | 74.67 | 77.51 | 87.50 | 77.48 |
| 16-shot | 54.22 | 90.96 | 81.62 | 82.53 | 85.95 | 79.26 | 80.97 | 91.07 | 80.82 |
| 32-shot | 61.04 | 92.84 | 84.35 | 84.12 | 87.23 | 81.64 | 83.04 | 93.81 | 83.51 |
| 64-shot | 61.95 | 93.48 | 85.76 | 84.70 | 88.64 | 83.80 | 84.28 | 95.58 | 84.77 |
| 128-shot | 63.92 | 93.53 | 87.04 | 85.74 | 89.51 | 84.96 | 85.31 | 96.20 | 85.78 |
| 256-shot | 64.90 | 93.59 | 88.02 | 86.56 | 89.93 | 85.06 | 86.08 | 97.05 | 86.40 |

*Table 4.* Performance comparison across different few-shot regimes.

| Method | Aircraft | Cars | CIFAR | CUB | Food | IN-R | SUN | UCF | Average |
|---|---|---|---|---|---|---|---|---|---|
| Imbalance | 48.25 | 90.61 | 83.46 | 82.15 | 86.58 | 84.23 | 84.09 | 92.29 | 81.46 |
| SCLIP$^-$ | 61.37 | 93.26 | 86.61 | 84.72 | 88.94 | 84.14 | 84.52 | 95.06 | 84.30 |

*Table 5.* Average accuracy $\bar{\mathcal{A}}$ of SCLIP$^-$ and Imbalance.

Sampling from a class-conditional Gaussian $\mathcal{N}(\mu_c, \Sigma_c)$ can be expressed as $\mathbf{x} = \mu_c + \mathbf{L}_c \mathbf{z}$, where $\mathbf{L}_c \mathbf{L}_c^\top = \Sigma_c$ is obtained via Cholesky decomposition. For each class, sampling requires a matrix decomposition with cost $\mathcal{O}(D^3)$ and feature generation with cost $\mathcal{O}(ND^2)$. As only class-wise means and covariances are stored, the decomposition is recomputed for each class at each incremental stage when synthetic features are generated. In our setting, the sampling budget ($N \in [16, 256]$) is significantly smaller than the feature dimensionality ($D = 512$), making the overall overhead dominated by the initial decomposition step.

Empirically, sampling 256 features per class achieves performance close to the joint-training upper bound, while incurring a generation-time cost comparable to much smaller budgets such as 16 samples. In contrast, the memory cost grows linearly with both the number of classes and generated samples, i.e., $\mathcal{O}(CND)$. Therefore, SCLIP$^-$ adopts a smaller sampling budget to substantially reduce memory consumption with only marginal performance degradation, making it particularly suitable for memory-constrained or large-scale incremental learning scenarios. By default, SCLIP employs a larger sampling budget when memory is not a limiting factor to better approximate the performance upper bound. Actually, in more resource-constrained scenarios, one may consider an even more economical alternative by directly storing the 16 synthesized features per class generated by SCLIP$^-$, thereby avoiding both the repeated matrix decomposition required for sampling and the memory overhead of maintaining class-wise statistics.

## D. Effect of Pre-Training Paradigms

Beyond CLIP-based visual encoders, we further examine whether the effectiveness of statistical feature modeling and replay extends across commonly used pre-training paradigms in class-incremental learning. In practice,

most recent CIL works adopt ViT backbones supervised-pretrained on ImageNet-1K or ImageNet-21K as default initialization. To assess the generality of our findings beyond this standard setting, we additionally consider representative self-supervised alternatives, including MAE (He et al., 2022b) and MoCo v3 (Chen et al., 2021). Specifically, we evaluate four widely used ViT-B/16 variants: supervised pre-training on ImageNet-1K and ImageNet-21K, as well as self-supervised pre-training via MAE and MoCo v3, all with frozen backbones and a trainable classification head.

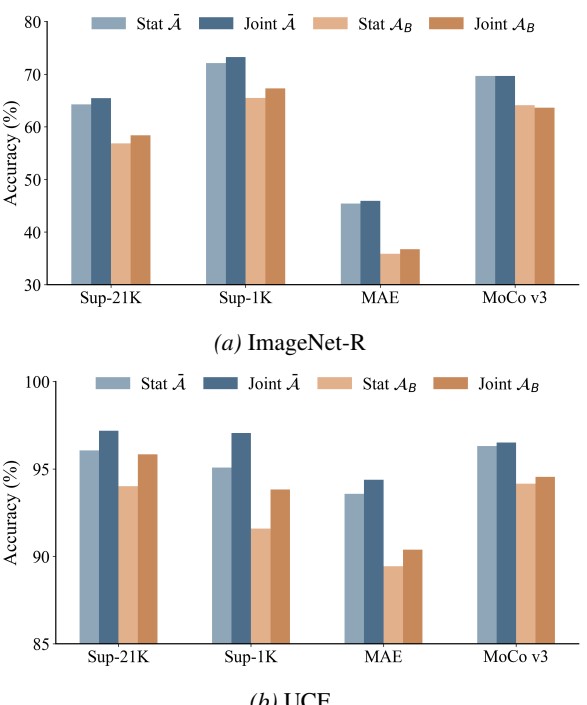

*(a)* ImageNet-R

*(b)* UCF

*Figure 10.* Statistical feature replay under different pre-training paradigms. Two representative metric average accuracy $\bar{\mathcal{A}}$ and final accuracy $\mathcal{A}_B$ are reported.

In contrast to CLIP-based models, these encoders operate purely in the visual domain and do not involve text embeddings or vision–language alignment. The classification head is a standard linear layer that maps the ViT feature dimension (768) to the number of classes. During incremental training, statistically generated visual features are fed into the classifier, which is optimized using a cross-entropy loss to reduce the discrepancy between predicted class probabilities and ground-truth labels. At inference time, a test image is passed through the frozen visual encoder and the trained classification head to obtain the final prediction.

For each pre-trained backbone, we compare an oracle joint-training setting, where all task data are jointly available, against a statistics-based replay setting, where class-wise Gaussian statistics are incrementally estimated and synthetic features sampled from these statistics are used to train the classifier. As summarized in Figure 10, across all pre-training paradigms, statistical replay consistently achieves performance close to the joint-training upper bound. While absolute performance varies across pre-training objectives, the relative gap between joint training and statistics-based replay remains small and stable, mirroring the behavior observed with CLIP visual encoders.

These results suggest that the effectiveness of SCLIP-style statistical replay is not tied to a specific pre-training objective, but instead arises from a more general property of strong pre-trained ViT representations: they induce well-structured, class-separable feature distributions that can be accurately summarized by low-order statistics.

As shown in Table 1, on ImageNet-R, moving from a uni-modal Gaussian to a GMM substantially improves held-out log-likelihood, revealing pronounced multi-modal structure in class-conditional feature distributions. This improvement translates into consistent accuracy gains over Mahalanobis-based inference across several benchmarks (Figure 4), most notably on ImageNet-R and SUN.

## E. Gaussian Feature Diagnostics

To better understand the statistical properties of class-wise feature distributions induced by strong pre-trained visual encoders, we report several diagnostic metrics in Table 1. This appendix provides precise definitions of these statistics and clarifies how they are computed.

**Gaussian Estimation.** For each class $c$, a single multivariate Gaussian distribution $\mathcal{N}(\mu_c, \Sigma_c)$ is estimated from feature vectors $g_i(\mathbf{x}) \in \mathbb{R}^d$, where $g_i(\cdot)$ denotes the frozen visual encoder and $d$ is the feature dimension. The mean $\mu_c$ and covariance $\Sigma_c$ are computed from a random 70% subset of features, with a small diagonal regularization added to $\Sigma_c$ for numerical stability. *For comparison, we additionally fit Gaussian mixture models (GMMs) on the same training*

*features using identical preprocessing.*

**Log-Likelihood (Log-Lik ↑).** Log-likelihood evaluates how well the estimated Gaussian fits held-out features from the same class. For each class $c$, we compute:

$$\mathcal{L}_c = \frac{1}{|\mathcal{D}_c|} \sum_{\mathbf{x} \in \mathcal{D}_c} \log \mathcal{N}\big(g_i(\mathbf{x}) \mid \mu_c, \Sigma_c\big), \qquad (9)$$

where $\mathcal{D}_c$ denotes the held-out subset of samples from class $c$. Higher values indicate better agreement between the Gaussian model and empirical features. In addition to the single Gaussian log-likelihood (G-LL), we also report log-likelihood under a Gaussian mixture model (GMM-LL), where the number of components is selected by BIC.

**Skewness (Skew ↓).** Skewness measures distribution asymmetry along each feature dimension. For a given class $c$, let $z_j$ denote the $j$-th feature dimension of $g_i(\mathbf{x})$, with mean $\mu_{c,j}$ and standard deviation $\sigma_{c,j}$. We compute the third standardized moment per dimension and report the mean absolute skewness:

$$\text{Skew}_c = \frac{1}{d} \sum_{j=1}^{d} \left| \mathbb{E}\left[ \left( \frac{z_j - \mu_{c,j}}{\sigma_{c,j}} \right)^3 \right] \right|. \qquad (10)$$

Lower values indicate more symmetric distributions.

**Kurtosis (Kurt ≈ 3).** Kurtosis measures tail heaviness of the feature distribution. Using the same notation as above, we compute the fourth standardized moment per dimension and average across dimensions:

$$\text{Kurt}_c = \frac{1}{d} \sum_{j=1}^{d} \mathbb{E}\left[ \left( \frac{z_j - \mu_{c,j}}{\sigma_{c,j}} \right)^4 \right]. \qquad (11)$$

A Gaussian distribution has kurtosis equal to 3.

**Silhouette Score (Silhouette ↑).** The silhouette score measures class separability in the feature space by comparing intra-class compactness and inter-class separation. Given a feature set $\mathcal{X} = \{g_i(\mathbf{x})\}$ with corresponding class labels, the silhouette score is computed as:

$$\text{Silhouette} = \frac{1}{|\mathcal{X}|} \sum_i \frac{b(i) - a(i)}{\max\{a(i), b(i)\}}, \qquad (12)$$

where $a(i)$ denotes the average distance between sample $i$ and other samples in the same class, and $b(i)$ denotes the minimum average distance between sample $i$ and samples from a different class. Higher values indicate better class separation.

## F. Compared Methods

In this section, we introduce the methods compared in the main paper. Note that all methods are re-implemented using

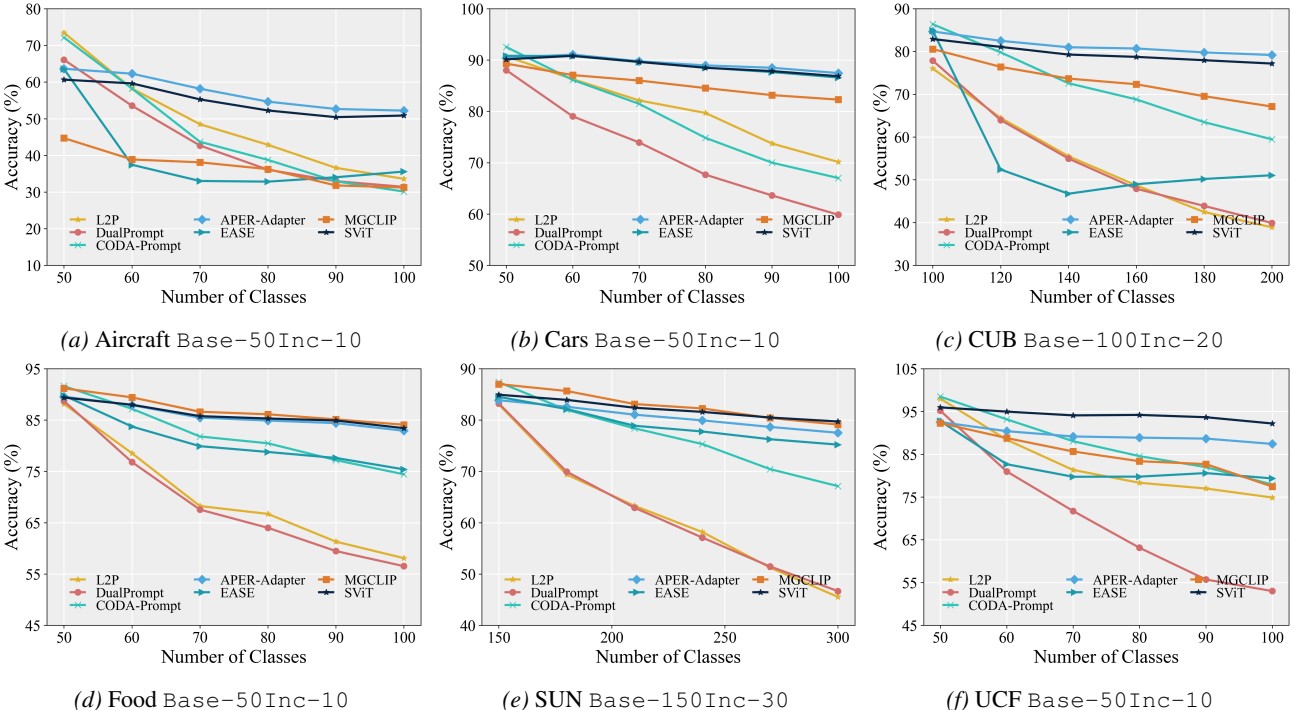

*Figure 11.* Incremental performance of different methods under `half-Base` settings.

the same pre-trained model for fairness: CLIP models are initialized with LAION-400M pre-trained weights, and for vision-only methods, the CLIP visual tower is used for initialization. Compared methods are listed as follows.

**L2P** (Wang et al., 2022b): constructs a pool of learnable prompts organized as key–value pairs, where the values guide task-specific representation learning and the keys are used to retrieve appropriate prompts.

**DualPrompt** (Wang et al., 2022a): explicitly decouples task-specific and task-shared prompts, and systematically investigates where and how different prompts should be attached to the backbone network.

**CODA-Prompt** (Smith et al., 2023): introduces an attention-based prompt composition mechanism to fuse multiple prompts, alleviating severe performance degradation caused by incorrect single-prompt selection.

**APER-Adapter** (Zhou et al., 2025a): is a *first-session* adaptation method that trains adapters only on the initial task. For subsequent tasks, it performs ensemble classification by combining class prototypes extracted from both the frozen pre-trained model and the adapted model, without further parameter updates.

**EASE** (Zhou et al., 2024): trains task-isolated adapters for each incremental task and performs joint decision-making by aggregating predictions from all task-specific adapters.

**RAPF** (Huang et al., 2024): leverages textual class representations to mitigate interference from newly introduced classes and employs an SVD-based parameter fusion module to enhance model robustness.

**PROOF** (Zhou et al., 2025c): learns an expandable projection space and utilizes self-attention to produce instance-specific embeddings for improved adaptability in continual learning.

**MGCLIP** (Huang et al., 2025): maintains a relatively stable modality gap while training a visual-space classifier that is not constrained by the modality alignment, enabling effective classification in continual settings.

## G. More Visualization Results

We further report detailed incremental performance curves under various base and incremental settings, as illustrated in Figure 11. SViT relies solely on the frozen visual encoder of CLIP and introduces no learnable parameters; as a result, it generally underperforms adaptive methods on the initial base tasks. This behavior is expected, since SViT does not explicitly optimize task-specific representations.

Notably, SViT exhibits more stable performance as the number of incremental classes increases. Across most datasets and protocols, its accuracy curves decay more gradually compared to other methods, leading to a competitive or even superior average accuracy over all tasks. This behav-

| Dataset | Task | 1 | 2 | 3 | 4 | 5 | 6 | 7 | 8 | 9 | 10 |
|---------|------|---|---|---|---|---|---|---|---|---|----|
| CUB | Classifier | 1.6 | 2.0 | 2.6 | 3.2 | 3.8 | 4.4 | 5.0 | 5.6 | 6.2 | 6.7 |
| | RAPF | 1.6 | 2.2 | 2.8 | 3.5 | 4.0 | 4.7 | 5.3 | 5.9 | 6.9 | 7.8 |
| | SViT | 1.8 | 2.3 | 2.9 | 3.6 | 4.3 | 5.1 | 5.7 | 6.4 | 7.3 | 8.2 |
| IN-R | Classifier | 2.6 | 4.2 | 5.7 | 7.1 | 8.2 | 9.9 | 11.1 | 13.1 | 14.1 | 15.4 |
| | RAPF | 2.5 | 4.4 | 6.6 | 9.5 | 9.6 | 11.6 | 13.8 | 15.3 | 17.3 | 19.6 |
| | SViT | 2.4 | 4.0 | 5.8 | 7.4 | 9.1 | 10.7 | 12.7 | 14.5 | 16.8 | 18.8 |
| SUN | Classifier | 8.6 | 15.5 | 19.8 | 24.7 | 28.4 | 34.7 | 38.6 | 45.0 | 49.0 | 53.7 |
| | RAPF | 8.4 | 12.8 | 20.2 | 27.1 | 31.8 | 38.1 | 44.2 | 51.0 | 54.6 | 63.1 |
| | SViT | 8.3 | 11.9 | 18.8 | 25.3 | 29.1 | 37.6 | 44.8 | 49.7 | 56.2 | 64.2 |
| UCF101 | Classifier | 1.3 | 1.9 | 2.6 | 3.1 | 3.8 | 4.5 | 5.1 | 5.7 | 6.4 | 7.0 |
| | RAPF | 1.4 | 2.3 | 3.7 | 4.7 | 4.7 | 5.5 | 5.9 | 6.9 | 7.6 | 8.4 |
| | SViT | 1.9 | 2.1 | 2.7 | 3.6 | 4.2 | 4.9 | 5.6 | 6.3 | 7.2 | 8.0 |

*Table 6.* Task-wise inference time comparison (in seconds).

ior indicates that the primary strength of SViT lies in its robustness to task progression and its ability to effectively mitigate catastrophic forgetting.

## H. Parameter Sensitivity Results

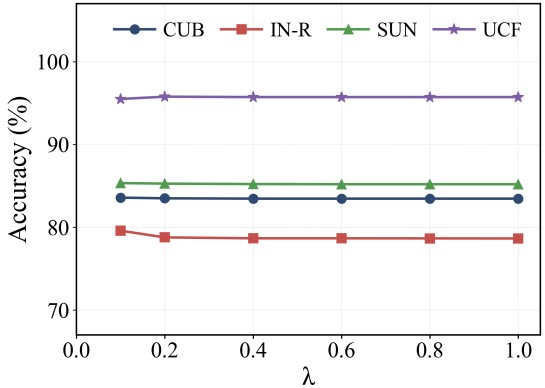

*Figure 12.* Parameter sensitivity.

We analyze the sensitivity of SViT to the fusion weight $\lambda$, which balances prototype-based similarity and GMM-based statistical scores. We vary $\lambda$ over a wide range while keeping all other settings fixed. As shown in Figure 12, SViT remains stable across different values of $\lambda$, indicating that the proposed fusion strategy is not sensitive to the precise choice of this hyperparameter.

## I. Running Time Results

We further analyze the computational overhead of SViT during both training and inference. Since SViT only instantiates class prototypes and lightweight GMM estima-

| Method | CUB | IN-R | SUN | UCF |
|--------|-----|------|-----|-----|
| RAPF | 396 | 939 | 2946 | 432 |
| PROOF | 1708 | 2579 | 6594 | 1025 |
| SViT | 136 | 257 | 1698 | 99 |

*Table 7.* Training time comparison (in seconds).

tors without continual optimization or replay-based training, its training cost remains substantially lower than existing replay-based approaches. As shown in Table 7, SViT consistently requires significantly less training time than RAPF and PROOF across all evaluated datasets.

We additionally investigate the inference overhead introduced by GMM-based fusion. Since SViT performs class-wise density evaluation during inference, its computational cost scales approximately linearly with the number of classes. To contextualize this overhead, we compare SViT with a standard unified linear classifier as well as RAPF. As shown in Table 6, the unified classifier consistently achieves the lowest latency, while RAPF and SViT exhibit similar scaling trends as the number of tasks increases.

Nevertheless, the latency gap between SViT and the standard classifier remains consistently small across datasets, suggesting that the practical inference overhead introduced by density-based fusion is minor. Even on larger benchmarks such as SUN and ImageNet-R, SViT maintains competitive inference efficiency.

