# OpenReview forum: "On the Power of Statistics in Class-Incremental Learning with Pretrained Models"
_ICML.cc/2026/Conference — ICML 2026 regular_

### Official Review · Reviewer_F2Sb · 2026-03-02

**Soundness:** 2
**Presentation:** 2
**Significance:** 3
**Originality:** 3
**Overall Recommendation:** 2
**Confidence:** 4

**Summary:**

This paper demonstrates that simple class-level feature statistics are a primary driver of success in class-incremental learning with pre-trained models.

**Compliance With Llm Reviewing Policy:**

Affirmed.

**Key Questions For Authors:**

Could you provide a formal comparison of the training time and inference between SVIT and the more complex CLIP-based methods like RAPF and PROOF?

**Limitations:**

Yes

**Strengths And Weaknesses:**

Strengths
1.	The theoretical analysis is rich and well-developed, offering meaningful support for the SviT framework.
2.	SviT is evaluated on multiple datasets, which enhances the credibility of the empirical results and highlights the robustness of the approach.

Weaknesses
1.	The main conclusions of the theoretical part are not sufficiently clear. A concise summary of the core contributions and findings would significantly improve the clarity and impact of the paper.
2.	Some empirical results that primarily serve to support the theoretical analysis could be moved to the appendix. This would improve the readability and flow of the main text while keeping the central narrative focused.
3.	The paper lacks sufficient explanations of the compared methods, such as L2P and CODA-Prompt, in the main text. Although descriptions are provided in the appendix, this is not sufficient for readers to fully understand the context and significance of the comparisons.
4.	It would be beneficial for the paper to include pseudocode for SviT and a detailed pseudocode description for SCLIP. This would improve reproducibility and help readers better understand the implementation details of the proposed methods.
5.	The metrics corresponding to Table 1 should be explicitly included in the paper. Without clearly defined evaluation metrics, the reported results remain difficult to interpret.
6.	Since SCLIP appears to be the most closely related method to generative replay approaches, the related work section should include a discussion of generative replay methods and their connection to the proposed framework. Additional related approaches should also be properly reviewed to situate the contribution within the broader literature.
7.	In Section 4.1, the theoretical assumptions require proper citations. Specifically:
a.	The assumption that class-conditional representations are approximately unimodal and can be reasonably modeled by a single Gaussian should be supported by relevant literature.
b.	The practice of maintaining class-level statistics and employing feature replay as an auxiliary mechanism to support learning across tasks should also be properly referenced.
8. The authors do not provide a formal comparison of time complexity or a detailed space complexity analysis beyond a high-level discussion in Section 5.2

---

> ### Author Rebuttal · Authors · 2026-03-28
>
> Thanks for the valuable feedback. We hope our responses address your concerns.
>
> **W1, W2, W4: improve the presentation**
>
> (1) We will distill core contributions as concisely as possible like: a) We empirically show that class-level statistics constitute an important and previously under-emphasized component of PTM-based CIL; b) Two simple yet effective baselines are explored to utilize class-level feature statistics and achieved good results on multiple benchmarks; c) We analyze the boundaries at which class-level statistics work and fail.
>
> (2) Part of the derivations of SViT will be moved to the appendix, while important theoretical analyses will be retained. This will result in the disappearance of Sections 4.2 and 4.3 to keep the central narrative focused.
>
> (3) Detailed pseudocode for SViT will be supplemented, and the training and inference process of SViT will be summarized after the definition of SViT.
>
> (4) We will revise Algorithm 1 in Appendix B to include a detailed description for SCLIP. Furthermore, codes and running logs will be made public.
>
> **W3, 6: discussion of related works**
>
> Thank you again for your constructive guidance.
>
> (1) For generative replay: We will revise the first subsection of Related Work to "Feature Replay in Class-Incremental Learning". The overall logic follows: a) The definition and categories of CIL; b) For feature replay, the most relevant concept in the broader CIL is generative replay. c) A discussion of generative and feature replay; d) Why we study feature replay.
>
> (2) For compared methods: We will revise the second subsection of Related Work and explain the context and significance of comparisons in the Experiment section and Appendix F. The second subsection of Related Work will describe representative methods of PTM-based CIL. The overall logic follows: a) PTM-based CIL mainly utilizes prompt, adapter, and LoRA fine-tuning. b) Describe how three technologies are applied to CIL and introduce the relevant methods, such as prompt-based L2P and CODA-Prompt, adapter-based APER-Adapter, and LoRA-based MGCLIP. c) We will point out that current high-performance methods are often accompanied by the use of class-level statistics, which are often regarded as implementation details and rarely analyzed.
>
> **W5, W7: metrics of Table 1, proper citations**
>
> Thank you for your careful review; this was our oversight. Table 2 reported average accuracy of $\bar{\mathcal{A}}$, which we will correct. Besides, we will add the appropriate citations after lines 126 and 129.
>
> **W8, K1: provide comparison of complexity**
>
> (1) Comparison of running time are listed as follows. During training, SViT only needs to instantiate two estimators, thus requiring significantly less time. We introduce a standard unified linear classifier to compare inference time. The linear classifier maintains relatively lower latency compared to SViT, while RAPF, shows a similar scaling trend. The gap between SViT and the classifier is consistently minor across datasets, suggesting that the practical impact on inference is negligible.
>
> **Table: Training time comparison.**
>
> | Method | CUB | IN-R | SUN | UCF|
> | :--- | :---: | :---: | :---: | :---: |
> | RAPF | 396 | 939 | 2946 | 432 |
> | PROOF | 1708 | 2579 | 6594 | 1025 |
> | SViT | 136 | 257 | 1698 | 99 |
>
> **Table: Task-wise inference time comparison.**
>
> | Dataset | Method | 1 | 2 | 3 | 4 | 5 | 6 | 7 | 8 | 9 | 10 |
> | :--- | :--- | :---: | :---: | :---: | :---: | :---: | :---: | :---: | :---: | :---: | :---: |
> | **CUB** | Classifier | 1.6 | 2.0 | 2.6 | 3.2 | 3.8 | 4.4 | 5.0 | 5.6 | 6.2 | 6.7 |
> | | RAPF | 1.6 | 2.2 | 2.8 | 3.5 | 4.0 | 4.7 | 5.3 | 5.9 | 6.9 | 7.8 |
> | | SViT | 1.8 | 2.3 | 2.9 | 3.6 | 4.3 | 5.1 | 5.7 | 6.4 | 7.3 | 8.2 |
> | **IN-R** | Classifier | 2.6 | 4.2 | 5.7 | 7.1 | 8.2 | 9.9 | 11.1 | 13.1 | 14.1 | 15.4 |
> | | RAPF | 2.5 | 4.4 | 6.6 | 9.5 | 9.6 | 11.6 | 13.8 | 15.3 | 17.3 | 19.6 |
> | | SViT | 2.4 | 4.0 | 5.8 | 7.4 | 9.1 | 10.7 | 12.7 | 14.5 | 16.8 | 18.8 |
> | **SUN** | Classifier | 8.6 | 15.5 | 19.8 | 24.7 | 28.4 | 34.7 | 38.6 | 45.0 | 49.0 | 53.7 |
> | | RAPF | 8.4 | 12.8 | 20.2 | 27.1 | 31.8 | 38.1 | 44.2 | 51.0 | 54.6 | 63.1 |
> | | SViT | 8.3 | 11.9 | 18.8 | 25.3 | 29.1 | 37.6 | 44.8 | 49.7 | 56.2 | 64.2 |
> | **UCF** | Classifier | 1.3 | 1.9 | 2.6 | 3.1 | 3.8 | 4.5 | 5.1 | 5.7 | 6.4 | 7.0 |
> | | RAPF | 1.4 | 2.3 | 3.7 | 4.7 | 4.7 | 5.5 | 5.9 | 6.9 | 7.6 | 8.4 |
> | | SViT | 1.9 | 2.1 | 2.7 | 3.6 | 4.2 | 4.9 | 5.6 | 6.3 | 7.2 | 8.0 |
>
> (2) Regarding space complexity: We provide a specific analysis of the large-scale dataset ImageNet-1K as: For ImageNet-1K, storing one image requires 224 x 224 x 3 = 150,528 bytes assuming integer storage. Storing one-class statistics requires ($512 + 512^2$) floats, i.e., 262,656 x 4 = 1,050,624 bytes. Therefore, storing one class-level statistics is equivalent to saving $1,050,624 \div 150,528 ≈ 7$ image instances. This is clearly far less than storing a standard example buffer of 20 samples per class.

---

> > ### Author Rebuttal · Reviewer_F2Sb · 2026-04-03
> >
> > Concerns cannot be resolved without another reviewing pass.

---

> > > ### Author Response · Authors · 2026-04-07
> > >
> > > Thank you for your constructive feedback and for acknowledging our rebuttal. We address each concern below.
> > >
> > > ---
> > >
> > > ### **W1: Summary of theoretical part**
> > >
> > > We agree that the main conclusions should be more explicit.
> > > We will add a new summary section (Section 4.5), titled **“Summary of the Power of Statistics”**, to consolidate the key findings:
> > >
> > > Through controlled analysis where class-level statistics are used as the sole supervision signal or directly for inference, we show that class-level statistics—particularly feature replay—can account for a substantial portion of class-incremental learning performance under strong pre-training.
> > >
> > > This leads to two implications:
> > >
> > > (1) **Diagnostic insight:** The contributions of many CIL methods may be confounded, as gains are often attributed to incremental strategies without isolating the role of the feature space and its statistical structure, highlighting the need for more controlled evaluation.
> > >
> > > (2) **Practical instantiation:** We introduce two minimal reference points—SCLIP (statistical replay) and SViT (replay-free inference)—to explicitly isolate and quantify the role of statistics, providing a simple way to assess when additional modeling is necessary.
> > >
> > > This reframing clarifies the role of class-level statistics and provides a clearer basis for distinguishing true continual learning gains from those induced by feature replay.
> > >
> > > ---
> > >
> > > ### **W2: Improve readability**
> > >
> > > To improve clarity and maintain a focused narrative:
> > >
> > > - Add a sentence after line 219 right column：See Appendix E for more empirical analysis of density modeling.
> > > - Move supporting analyses (e.g., Table 1, Figure 5, and discussions) to **Appendix E**
> > > - Reorganize Appendix E as **“Empirical Analysis of the Limitations of Density Modeling”**, including:
> > >   - **E.1:** Empirical and qualitative analysis (moved from the main text)
> > >   - **E.2:** Diagnostic metrics (original Appendix E content)
> > >
> > > - In **E.1**, structure the discussion into:
> > >   - (a) Mahalanobis behavior differences
> > >   - (b) GMM modeling analysis
> > >
> > > The main text will retain only key insights, improving clarity and flow.
> > >
> > > ---
> > >
> > > ### **W3: Explain comparisons**
> > >
> > > We agree that the main text lacks sufficient explanation of the compared methods.
> > >
> > > Section 5.1 will be revised to include concise descriptions:
> > >
> > > - For **SCLIP comparisons**:
> > >   - RAPF: feature replay using synthetic features
> > >   - PROOF: exemplar replay
> > >
> > > - For **SViT comparisons**, representative PTM-based CIL methods:
> > >   - Prompt-based: L2P, DualPrompt, CODA-Prompt
> > >   - Adapter-based: APER-Adapter, EASE
> > >   - LoRA-based: MGCLIP
> > >
> > > Each method will be briefly described by its adaptation mechanism, making the comparisons self-contained.
> > >
> > > ---
> > >
> > > ### **W4: Pseudocode and algorithm clarity**
> > >
> > > We will improve reproducibility and clarity:
> > >
> > > - Algorithm tables for SCLIP and SViT are available at:
> > >   https://anonymous.4open.science/r/SViT-44C2/README.md
> > >
> > > - We will incorporate **step-by-step pseudocode and structured summaries**:
> > >
> > >   - **SViT**:
> > >     - Add step-by-step pseudocode
> > >     - Include a **“Summary of SViT”** covering the full pipeline: statistics → prototype → GMM → similarity–density fusion
> > >
> > >   - **SCLIP**:
> > >     - Extend Algorithm 1 in Appendix B with full pseudocode (statistics update, feature sampling, adapter training)
> > >     - Add a **“Summary of SCLIP”** to organize both training and inference procedures
> > >
> > > This combination improves clarity, reproducibility, and ease of implementation.
> > >
> > > ---
> > >
> > > ### **W6: Discussion of generative replay**
> > >
> > > We will revise the Related Work section to better contextualize our method:
> > >
> > > - Remove line 87 (“SCLIP samples features … Appendix A”)
> > > - Add a subsection: **Feature Replay under Strong Pre-training**, incorporating a concise version of Appendix A
> > >
> > > Specifically:
> > >
> > > - Briefly review **generative replay**
> > > - Introduce **statistical feature replay**
> > > - Clarify that **SCLIP can be viewed as a simplified feature-space instantiation of generative replay**, strengthening its connection to replay-based approaches
> > >
> > > ---
> > >
> > > ### **Other comments**
> > >
> > > **W5 (metrics):**
> > > The metrics in Table 1, including their definitions and computation details, are provided in Appendix E (Page 15). Each column in Table 1 corresponds to a metric defined in Appendix E.
> > >
> > > **W7 (citations):**
> > > The citations of W7.a are Zhang et al., 2023; Wang et al., 2023a in the References.
> > > The citations of W7.b are Sun et al., 2025; Wang et al., 2025 in the References.
> > > We will revise the paper to include these proper citations.
> > >
> > > **W8 (complexity):**
> > > Detailed running time comparisons and space analysis have been provided in the rebuttal, and will be incorporated into the paper as an additional appendix: “Running Time Comparison”.
> > >
> > > ---
> > >
> > > We thank the reviewer again for the insightful suggestions. We believe these changes will improve the clarity and positioning of the paper. We kindly ask the reviewer to consider raising the overall score to reflect the fully resolved concerns.

---

### Official Review · Reviewer_tB6Y · 2026-03-09

**Soundness:** 3
**Presentation:** 2
**Significance:** 3
**Originality:** 3
**Overall Recommendation:** 5
**Confidence:** 4

**Summary:**

This paper investigates the role of class-level feature statistics in pre-trained model-based class-incremental learning (CIL). By proposing two minimal baselines, SCLIP (statistics-driven replay) and SVIT (training-free statistical inference), the authors demonstrate that simple statistical estimators can effectively recover performance comparable to static joint training.

**Compliance With Llm Reviewing Policy:**

Affirmed.

**Final Justification:**

The authors have addrssed my concerns, and I raise my score.

**Key Questions For Authors:**

See weaknesses.

**Limitations:**

Yes.

**Strengths And Weaknesses:**

Strengths

1.	This paper reveals through the minimal SCLIP and SVIT baselines that class-level feature statistics are the true core of maintaining model performance.
2.	The experimental evaluation is comprehensive. The paper conducts extensive experiments across eight diverse benchmark datasets, comparing the proposed baselines against a wide range of recent state-of-the-art methods. Furthermore, the inclusion of detailed analyses on memory cost, parameter sensitivity provides a highly thorough and convincing empirical validation of the core claims.

Weaknesses

1.	Although the paper ultimately proposes SVIT as its core, training-free, and replay-free solution for class-Incremental Learning, it dedicates a disproportionately large amount of space in the methodology section to discussing SCLIP. This structural imbalance confuses the reader and blurs the primary contribution of the paper.
2.	Figure 4 shows that GMM performs similarly or better to the final SVIT results. Why not direcly use GMM? Could you provide exact numerical tables or significance tests corresponding to the results in Figure 4? This is necessary to prove whether SVIT's fusion strategy actually yields statistically significant gains over the pure GMM baseline.
3.	The paper notes that SVIT struggles on datasets like ImageNet-R due to substantial distribution overlap and domain gap. How to address this problem?

---

> ### Author Rebuttal · Authors · 2026-03-28
>
> Thanks for the valuable feedback. We hope our responses address your concerns.
>
> **W1: structural imbalance**
>
> We sincerely apologize for the confusion caused by the structural imbalance. The core of this paper is to demonstrate that good class-incremental learning can be achieved using only class-level feature statistics without complex design. A crucial reference point is SCLIP, used to dissect the underestimation of the role of class-level statistics in current methods. Another reference point is SViT, which fully utilizes class-level statistics directly for inference. These two reference points investigate two scenarios of continual learning: a leniently replay-allowed scenario and a very strict one. Given that SCLIP may be the minimal reconstruction obtained by decoupling existing methods, and SViT is a powerful new survey that utilizes statistics without training or replay, we will reorganize the structure, reduce the space of SCLIP, and move it to the appendix. Besides, to highlight the paper's intrinsic contributions, we will restate the beginning of Section 4 and maintain the central narrative focus on SViT.
>
> **W2: Why SViT**
>
> (1) We provide SViT with lambda set to 0.1, and the experimental results is the same as in Figure 4 of the paper, as shown in the table below. In addition, lambda for all experiments in the main paper is set to 1, and we will add these numerical results to the appendix later.
>
> **Table: Numerical results of Figure 4.**
>
> | Method \ Dataset | Aircraft | Cars | CIFAR | CUB | Food | IN-R | SUN | UCF |
> | :--- | :---: | :---: | :---: | :---: | :---: | :---: | :---: | :---: |
> | **Prototype** | 59.13 | 92.10 | 84.18 | 83.83 | 87.92 | 81.07 | 82.22 | 90.37 |
> | **Mahalanobis** | 65.74 | 92.18 | 86.16 | 80.60 | 87.44 | 51.62 | 69.63 | 99.02 |
> | **GMM** | 61.13 | 91.89 | 85.96 | 83.38 | 88.86 | 78.62 | 85.19 | 95.72 |
> | **SViT** | 61.33 | 91.94 | 86.13 | 83.58 | 88.97 | 79.59 | 85.34 | 95.49 |
>
> (2) The main reason we retain the prototype and integrate it is that when facing datasets like ImageNetR with substantial distribution overlap, we can degrade to a prototype-leading method, in which case SViT retains its global discriminative ability. In the "lower lambda" table, we further modified $\lambda$. The results show that ImageNet-R achieves better results while other datasets exhibit negligible fluctuations as SViT gradually degrades as a prototype-leading classifier.
>
> **Table: Lower lambda.**
>
> | $\lambda$ | CUB $\bar{\mathcal{A}}$ | IN-R $\bar{\mathcal{A}}$ | SUN $\bar{\mathcal{A}}$ | UCF $\bar{\mathcal{A}}$ |
> | :--- | :--- | :--- | :--- | :--- |
> | 0.1 | **83.58** | **79.59** | **85.34** | 95.49 |
> | 0.15 | 83.50 | 78.93 | 85.31 | **95.80** |
> | 0.20 | 83.50 | 78.78 | 85.27 | 95.77 |
>
> (3) On datasets like Food, SUN, and UCF, richer class-level information leads to better performance, as reflected in the significant difference between SViT and APER-Adapter in Table 3. Therefore, SViT combines the advantages of global semantic alignment and local distribution cues.
>
> **W3: struggles on datasets like ImageNet-R**
>
> SViT directly processes datasets like ImageNet-R by degrading the combined classifier to a prototype-leading classifier. The reasoning can be found in W2's response (2). Besides, it is worth noting that SViT was designed to directly investigate the influence of class-level information under the most stringent conditions, thus sacrificing some “plasticity” to improve performance on ImageNetR. However, this is an acceptable trade-off; while we could certainly perform adaptive training in the first-session like APER, this would also compromise the pre-trained knowledge. Aside from this, SViT achieved good results on multiple benchmarks.

---

> > ### Author Rebuttal · Reviewer_tB6Y · 2026-04-01
> >
> > My concerns have been adequately addressed. The new experiments and clarifications improved the quality of the manuscript. I would like to raise my score to Accept.

---

> > > ### Author Response · Authors · 2026-04-01
> > >
> > > Thank you for your constructive feedback and for acknowledging our rebuttal. We appreciate your thoughtful review and are glad that our clarifications addressed your concerns.

---

### Official Review · Reviewer_LGx8 · 2026-03-11

**Soundness:** 3
**Presentation:** 3
**Significance:** 3
**Originality:** 3
**Overall Recommendation:** 6
**Confidence:** 5

**Summary:**

This paper explores the simple class-level feature statistics in incremental class learning (CIL) based on powerful pre-trained visual models. Using multiple benchmark datasets, it is demonstrated that storing and incrementally updating compact class-level statistics, rather than continuously updating model parameters, can recover most of the performance typically achieved by more complex state-of-the-art CIL methods.

**Compliance With Llm Reviewing Policy:**

Affirmed.

**Final Justification:**

This is an interesting work investigating the power of class-level statistics in class incremental learning. I have also read other comments and feedback, and I personally believe this work makes a significant contribution, especially SCLIP, which directly validates that simple feature replay achieves better results by removing complex design modules. This will be very valuable for future community development, considering that subsequent methods must consider whether feature replay is the dominant factor in performance gain and design methods that are truly resistant to forgetting. Therefore, I'd like to champion the acceptance for this work.

**Key Questions For Authors:**

1. How exactly were RAPF/MOS modified to remove statistics-driven feature replay? Since Figure 1 underpins the central claim, more explicit ablation details are necessary for reproducibility.

2. Table 2 compares SCLIP mainly against RAPF/PROOF; it would be more convincing to include the broader baselines from Table 3 or a justification showing why RAPF/PROOF are sufficient proxies.

**Limitations:**

Yes

**Strengths And Weaknesses:**

Strengths:

1. This paper carries out a comprehensive and systematic experimental investigation of the importance of class-level feature statistics in contemporary CIL approaches. It demonstrates the substantial effect these statistics have on practical outcomes.

2. The paper presents two crucial benchmarks, SCLIP and SViT, in an accessible manner and provides a clear visualization of the complete procedure.

3. The manuscript is well-structured, articulate, and openly addresses its results and constraints.

4. The experimental section is fairly comprehensive, employing eight benchmarks and diverse CIL protocols.


Weakness:

1. Some methodological specifics are not thoroughly explained. Equation (4) provides a standard definition of the Mahalanobis distance, but the classification decision rule is not explicitly stated; it must be inferred that the class with the smallest d^Maha is chosen.

2. Figure 8 contains a high density of information, and a central component is its analysis of how the number of real samples used for estimating class-wise statistics impacts results. This key insight shows that performance can decline under extreme few-shot conditions. Consequently, it highlights scenarios where straightforward statistical summarization fails, making it essential to discuss this in the main text.

---

> ### Author Rebuttal · Authors · 2026-03-28
>
> Thanks for the valuable feedback. We hope our responses address your concerns.
>
> **W1: methodological specifics**
>
> Thank you for your careful review. We will add a formal explanation of the classification decision rule after Equation (4).
>
> **W2: Discuss scenarios where statistical summarization fails**
>
> Thank you for your constructive suggestions. Discussions of some of the most relevant and important variants of SCLIP will be added to the end of Section 4.1, such as exploring when statistics are effective for k-shots and prototype-based methods.
>
> **Q1: How to remove feature replay**
>
> MOS is heavily based on repo [PILOT](https://github.com/LAMDA-CL/LAMDA-PILOT "PILOT"); we can remove statistics-driven feature replay by simply setting the parameter "crct_epochs" in the JSON-based configuration to 0. For RAPF, all modules except parameter fusion involve feature replay; therefore, we only need to retain the training of in-task samples and the parameter fusion module of the adapter to remove statistics-driven feature replay.
>
> **Q2: Why RAPF/PROOF are sufficient**
>
> Thank you for your valuable suggestion. In fact, SCLIP has already been compared with the upper bound of joint training in Section 4.1, which is sufficient to illustrate the effectiveness of SCLIP. Therefore, no further methods are needed to verify the effect of SCLIP. We will add the upper bound to Table 2 for a more intuitive explanation.

---

> > ### Author Rebuttal · Reviewer_LGx8 · 2026-04-03
> >
> > The rebuttal has partially addressed the my concerns in my original review. Considering the details of "formal explanation of the classification decision rule", "the most relevant and important variants of SCLIP" and "the upper bound to Table 2" haven't been illustrated in the response, I still encourage the authors to explicitly discuss the details and to calibrate the novelty phrasing carefully.

---

> > > ### Author Response · Authors · 2026-04-03
> > >
> > > Thank you for your valuable feedback. We hope the following responses address your concerns.
> > >
> > > **W1: methodological specifics**
> > >
> > > Under the Gaussian distribution assumption, Mahalanobis distance is equivalent to the negative log-likelihood measure of a sample relative to its class. Therefore, when we have Mahalanobis distances between a sample and different classes, we choose the class with the smallest distance as the classification result. We will add a formal explanation of the classification decision rule after Equation (4): "Classification is performed by assigning each sample to the class with the smallest Mahalanobis distance."
> > >
> > > **W2: Discuss the most relevant variants**
> > >
> > > (1) We will discuss the most relevant variants of SCLIP in Section 4.1 as follows. First, we will replace Figure 2 with Figure 8 from Appendix B. Considering that the content of Figure 2 could be presented in more detail in Table 2 (refer to **Q2**, we will move the specific values from Figure 2 to Table 2), we will delete Figure 2.
> > >
> > > (2) The new Figure 2 (originally Figure 8 in Appendix B) discusses the SCLIP variants $k$-shot, $Prototype$, and $Variance$. In the main paper, we need to include $k$-shot and $Prototype$ in the discussion. One proves the conditions under which SCLIP holds, and the other proves why SViT is superior to Prototype-based methods. Some specific implementation details, definitions of variants, and more detailed discussions of variants will be kept in Appendix B to keep the main text concise. Therefore, we will add the following specific description at the end of Section 4.1 after Line 199: Besides, Figure 2 also explores the number of samples ($k$-shot) required to estimate class-level statistics; a moderate number of samples is sufficient to recover most of the performance achieved through joint training. This figure further demonstrates that first-order information ($Prototype$) alone is insufficient to describe class-level information and support better performance.
> > >
> > > (3) Considering that the new Figure 2 (originally Figure 8 in Appendix B) highly compresses the information in $k$-shot, we will add the following table to Appendix B. When Appendix B discusses $k$-shot in more detail, following line 703, it will specify the exact number of samples required for a single dataset to prevent Gaussian estimation from collapsing. We will add the following statement: In addition, to prevent Gaussian estimation from collapsing, we provide the minimum number of samples required for a single dataset in Figure x (a new table shown below). From Figure x, we can conclude that for most datasets, only 32 samples per class are sufficient to obtain good results, while more difficult datasets such as Aircraft and ImageNet-R may require more samples, such as 64 samples per class, to achieve competitive results.
> > >
> > > | Method | Aircraft | Cars | CIFAR | CUB | Food | IN-R | SUN | UCF | Average |
> > > | :--- | :---: | :---: | :---: | :---: | :---: | :---: | :---: | :---: | :---: |
> > > | 8-shot | 49.68 | 90.19 | 79.04 | 78.36 | 82.91 | 74.67 | 77.51 | 87.50 | 77.48 |
> > > | 16-shot | 54.22 | 90.96 | 81.62 | 82.53 | 85.95 | 79.26 | 80.97 | 91.07 | 80.82 |
> > > | 32-shot | 61.04 | 92.84 | 84.35 | 84.12 | 87.23 | 81.64 | 83.04 | 93.81 | 83.51 |
> > > | 64-shot | 61.95 | 93.48 | 85.76 | 84.70 | 88.64 | 83.80 | 84.28 | 95.58 | 84.77 |
> > > | 128-shot | 63.92 | 93.53 | 87.04 | 85.74 | 89.51 | 84.96 | 85.31 | 96.20 | 85.78 |
> > > | 256-shot | 64.90 | 93.59 | 88.02 | 86.56 | 89.93 | 85.06 | 86.08 | 97.05 | 86.40 |
> > >
> > > **More about Q1**
> > >
> > > Regarding MOS, we can simply set the parameter crct_epochs to 0. Regarding RAPF, we can remove $L$-hinge and set the parameter memory_data to None. All code and running logs will be released at GitHub after the acceptance.
> > >
> > > **Q2: the upper bound to Table 2**
> > >
> > > The specific values for Upper-Bound will be added to Table 2 as shown below, the table reports the average accuracy $\bar{\mathcal{A}}$ on multiple benchmarks. The optimal result excluding Upper-Bound is highlighted in bold. When SCLIP uses only feature replay while Upper-Bound uses all task data, the performance difference between SCLIP and Upper-Bound is almost negligible, and SCLIP is significantly better than RAPF and PROOF. Considering that Upper-Bound directly quantifies performance under ideal conditions, SCLIP does not require other methods to prove its effectiveness.
> > >
> > > | Method | Aircraft | Cars | CIFAR | CUB | Food | IN-R | SUN | UCF | Average |
> > > | :--- | :---: | :---: | :---: | :---: | :---: | :---: | :---: | :---: | :---: |
> > > | Upper-Bound | 65.40 | 93.41 | 89.16 | 85.71 | 91.01 | 85.45 | 87.58 | 97.97 | 86.96 |
> > > | RAPF | 59.28 | 92.58 | 87.47 | 84.24 | 89.48 | 85.11 | 85.95 | 95.05 | 84.49 |
> > > | PROOF | 64.61 | 92.70 | 86.77 | 84.87 | 90.04 | 84.19 | 83.75 | 94.34 | 84.16 |
> > > | SCLIP$^{-}$ | 61.37 | 93.26 | 86.61 | 84.72 | 88.94 | 84.14 | 84.52 | 95.06 | 84.30 |
> > > | SCLIP | **65.67** | **93.69** | **88.64** | **86.53** | **90.54** | **85.85** | **86.74** | **97.22** | **86.86** |

---

### Official Review · Reviewer_XG5N · 2026-03-14

**Soundness:** 3
**Presentation:** 3
**Significance:** 2
**Originality:** 2
**Overall Recommendation:** 4
**Confidence:** 4

**Summary:**

This paper investigates the underlying reasons for the success of recent Class-Incremental Learning (CIL) methods that utilize large-scale Pre-trained Models (PTMs). The authors hypothesize that class-level feature statistics are the primary driver of performance in these settings. To test this, they introduce two minimal reference points: SCLIP, a replay-based method that uses synthetic features sampled from class-wise Gaussian distributions , and SVIT, a training-free inference method that combines prototype similarity with Gaussian Mixture Model (GMM) density estimation. Their experiments across eight benchmarks show that these simple statistical approaches can recover a significant portion of the performance achieved by far more complex state-of-the-art CIL algorithms.

**Compliance With Llm Reviewing Policy:**

Affirmed.

**Final Justification:**

After considering the strong support from other reviewers and the value of the provided baselines, I am persuaded that the insights regarding class-level statistics outweigh my initial concerns about technical novelty; accordingly, I am raising my score.

**Key Questions For Authors:**

1. **Technical Novelty:** What is the specific delta over PTM-based CIL methods like SLCA or RAPF? Beyond confirming the utility of statistics, does the paper identify a unique failure mode of prior art that SVIT specifically addresses?
2. **Memory Scalability:** How does the $O(D^2)$ storage cost for covariance matrices compare to standard exemplar buffers in large-scale settings (e.g., ImageNet-1K)? Please provide a quantitative break-even analysis.
3. **Robustness and Sample Efficiency:** How does the model handle inherently ambiguous features in distribution-shift scenarios like ImageNet-R? Additionally, what is the minimum number of samples required to prevent Gaussian estimation from collapsing?
4. **Inference Latency:** Does the inference overhead of GMM-based fusion scale linearly with tasks, and how does its latency compare to the constant-time inference of a standard unified linear classifier?

**Limitations:**

Yes

**Strengths And Weaknesses:**

### **Strengths**

- **Insightful Empirical Analysis**: The paper provides a valuable "reality check" by demonstrating that many complex architectural innovations in PTM-based CIL may be secondary to simple statistical modeling.

- **Comprehensive Benchmarking**: The authors evaluate their claims across diverse datasets and various pre-training paradigms, including supervised and self-supervised ViT models.

- **Clear Minimal Baselines**: SCLIP and SVIT serve as highly interpretable and effective reference points for future CIL research.

- **Stable Performance**: The proposed SVIT method shows remarkable stability against task progression and is robust to hyperparameter variations in its fusion weight $\lambda$.


---
### **Weaknesses**

1. **Limited Methodological Innovation**
The paper is primarily a post-hoc analysis rather than a proposal for a novel, high-performance algorithm. While the observation that statistics are powerful is important, the techniques used, such as Gaussian feature replay and GMMs, are well-established in the literature. Consequently, the paper offers limited new technical tools for the community to solve existing CIL challenges.


2. **Over-reliance on Strong PTM Representations**
The effectiveness of the proposed statistical approach is strictly tied to the quality of the frozen backbone. As the authors note, these properties may not hold for weaker backbones or in scenarios where the feature space is not naturally well-structured. This makes the conclusion somewhat narrow, as it characterizes the power of the *pre-training* rather than a breakthrough in *incremental learning* itself.


3. **Failure in High-Overlap and Complex Scenarios**
The methods struggle significantly on challenging datasets like ImageNet-R, where class distributions overlap or exhibit high intra-class variability. Because SVIT is training-free, it lacks the "plasticity" required to refine representations or decision boundaries in these ambiguous regions. Furthermore, the assumption of Gaussian distributions remains a fundamental bottleneck that the paper acknowledges but does not fundamentally resolve.

---

> ### Author Rebuttal · Authors · 2026-03-28
>
> Thanks for the valuable feedback. We hope our responses address your concerns.
>
> **W1, Q1: Limited Innovation**
>
> (1) The core of this paper is: simple class-level statistics achieve strong performance, rather than offering new technical tools. We believe this finding is equally valuable because feature replay needs to be considered in future algorithms to be wary of designing more efficient modules, rather than the inherent performance likely stemming from the applied feature replay. In addition, based on the fact that statistics are powerful, we have introduced two simple yet effective methods.
>
> (2) Compared to SLCA and RAPF, SCLIP directly quantifies the ability of feature replay rather than being an underestimated auxiliary technique.
>
> (3) The failure mode that SViT addresses is that APER-Adapter only utilize first-order information. Even though SViT doesn't adapt learning parameters, SViT still shows significant advantages over APER-Adapter on Food, SUN, and UCF (see Table 3). As stated in line 695, prototype-based methods are clearly insufficient to describe class-level information.
>
> **W2: Over-reliance on Strong PTM**
>
> We sincerely appreciate the reviewer's insightful feedback. Recent advances have almost focused on PTM settings, which is the starting point of our approach, not a strict limitation. We will carefully revise the conclusion to ensure not too narrow.
>
> **W3, Q3: Failure in Complex Scenarios**
>
> (1) SViT was designed to verify the classification performance achievable solely through class-level statistics under the most stringent conditions. While we could certainly perform a first-session adaptation like APER, this would also compromise the pre-trained knowledge.
>
> (2) SViT directly processes datasets like ImageNet-R by degrading to a prototype-leading classifier. In the following table, we further reduce $\lambda$, showing that ImageNet-R achieves better results when other datasets show negligible changes.
>
> (3) Regarding the Gaussian assumption: Section 4.3 specifically discusses this constraint and introduces GMM to transcend the single-modality assumption. While more complex distribution modeling can improve likelihood, combining global semantic alignment is more effective than pursuing complex probabilistic modeling.
>
> | $\lambda$ | CUB $\bar{\mathcal{A}}$ | IN-R $\bar{\mathcal{A}}$ | SUN $\bar{\mathcal{A}}$ | UCF $\bar{\mathcal{A}}$ |
> | :--- | :--- | :--- | :--- | :--- |
> | 0.1 | **83.58** | **79.59** | **85.34** | 95.49 |
> | 0.15 | 83.50 | 78.93 | 85.31 | **95.80** |
> | 0.20 | 83.50 | 78.78 | 85.27 | 95.77 |
>
> **Q2: Memory Scalability**
>
> For ImageNet-1K, storing one image requires 224 x 224 x 3 = 150,528 bytes assuming integer storage. Storing the statistics for one class requires ($512 + 512^2$) float parameters, i.e., 262,656 x 4 = 1,050,624 bytes. Therefore, storing one class-level statistics is equivalent to saving $1,050,624 \div 150,528 ≈ 7$ image instances. However, a standard exemplar buffer stores 20 samples per class.
>
> **Q3: Sample Efficiency**
>
> We list the performance in the table below. Overall, 32 samples per class are sufficient to achieve good results. For challenging datasets, such as Aircraft and ImageNetR, more samples are required.
>
> | Method | Aircraft | Cars | CIFAR | CUB | Food | IN-R | SUN | UCF | Average |
> | :--- | :---: | :---: | :---: | :---: | :---: | :---: | :---: | :---: | :---: |
> | 8-shot | 49.68 | 90.19 | 79.04 | 78.36 | 82.91 | 74.67 | 77.51 | 87.50 | 77.48 |
> | 16-shot | 54.22 | 90.96 | 81.62 | 82.53 | 85.95 | 79.26 | 80.97 | 91.07 | 80.82 |
> | 32-shot | 61.04 | 92.84 | 84.35 | 84.12 | 87.23 | 81.64 | 83.04 | 93.81 | 83.51 |
> | 64-shot | 61.95 | 93.48 | 85.76 | 84.70 | 88.64 | 83.80 | 84.28 | 95.58 | 84.77 |
> | 128-shot | 63.92 | 93.53 | 87.04 | 85.74 | 89.51 | 84.96 | 85.31 | 96.20 | 85.78 |
> | 256-shot | 64.90 | 93.59 | 88.02 | 86.56 | 89.93 | 85.06 | 86.08 | 97.05 | 86.40 |
>
> **Q4: Inference Latency**
>
> The inference overhead scale linearly with classes. As shown in the table, the unified linear classifier maintains relatively lower latency. However, the gap is consistently minor across datasets, suggesting that the practical impact is negligible.
>
> | Dataset | Method | 1 | 2 | 3 | 4 | 5 | 6 | 7 | 8 | 9 | 10 |
> | :--- | :--- | :---: | :---: | :---: | :---: | :---: | :---: | :---: | :---: | :---: | :---: |
> | **CUB** | Classifier | 1.6 | 2.0 | 2.6 | 3.2 | 3.8 | 4.4 | 5.0 | 5.6 | 6.2 | 6.7 |
> | | SViT | 1.8 | 2.3 | 2.9 | 3.6 | 4.3 | 5.1 | 5.7 | 6.4 | 7.3 | 8.2 |
> | **IN-R** | Classifier | 2.6 | 4.2 | 5.7 | 7.1 | 8.2 | 9.9 | 11.1 | 13.1 | 14.1 | 15.4 |
> | | SViT | 2.4 | 4.0 | 5.8 | 7.4 | 9.1 | 10.7 | 12.7 | 14.5 | 16.8 | 18.8 |
> | **SUN** | Classifier | 8.6 | 15.5 | 19.8 | 24.7 | 28.4 | 34.7 | 38.6 | 45.0 | 49.0 | 53.7 |
> | | SViT | 8.3 | 11.9 | 18.8 | 25.3 | 29.1 | 37.6 | 44.8 | 49.7 | 56.2 | 64.2 |
> | **UCF** | Classifier | 1.3 | 1.9 | 2.6 | 3.1 | 3.8 | 4.5 | 5.1 | 5.7 | 6.4 | 7.0 |
> | | SViT | 1.9 | 2.1 | 2.7 | 3.6 | 4.2 | 4.9 | 5.6 | 6.3 | 7.2 | 8.0 |

---

> > ### Author Rebuttal · Reviewer_XG5N · 2026-04-03
> >
> > The rebuttal addresses my technical inquiries with helpful quantitative data on memory scalability, sample efficiency, and latency . However, methodological novelty remains limited, as the techniques used are well-established in the literature . Furthermore, the reliance on a prototype-based approach for complex cases like ImageNet-R suggests that the proposed statistical modeling lacks the inherent robustness to handle significant distribution overlap . As the results characterize the strength of pre-trained representations rather than a fundamental advancement in incremental learning , I maintain my score.

---

> > > ### Author Response · Authors · 2026-04-05
> > >
> > > Thank you for your constructive feedback and for acknowledging our rebuttal. We hope the following clarifications address your concerns.
> > >
> > > ---
> > >
> > > **W1: Limited novelty**
> > >
> > > We thank the reviewer for raising the concern regarding methodological novelty. We agree that the techniques used (e.g., feature replay, GMM) are well-established. However, our contribution lies **not in introducing new components**, but in **isolating and quantifying their role to answer a previously unclear question**:
> > >
> > > > **How much of the performance in PTM-based CIL can be attributed solely to class-level statistics?**
> > >
> > > Our key contribution is to transform class-level statistics from an *implicit auxiliary mechanism* into an *explicit object of study*. In prior work, feature replay is tightly coupled with other modules (e.g., adapters or fusion), making it unclear whether gains arise from architecture or statistical effects. By removing this entanglement, we show that class-level statistics alone account for a substantial portion of performance.
> > >
> > > To support this, we construct two simple instantiations:
> > > - **SCLIP**, showing that statistical replay with lightweight adaptation achieves strong performance;
> > > - **SViT**, showing that even without parameter updates, class-level statistics can directly support inference.
> > >
> > > These results indicate that feature replay is not merely an implementation detail, but a dominant factor in many existing methods. Thus, while the building blocks are not novel, the **explicit isolation, quantitative evaluation, and validation of class-level statistics** provide a meaningful and underexplored perspective for PTM-based CIL.
> > >
> > > ---
> > >
> > > **W2: About Strong PTM**
> > >
> > > We appreciate the reviewer’s concern that our results may mainly reflect the strength of pre-trained representations. While PTMs are indeed crucial, we argue that our contribution goes beyond this.
> > >
> > > In modern CIL, **the contribution of PTMs and incremental strategies is often entangled**, leading to gains being attributed to new modules without understanding the role of the feature space.
> > >
> > > Our work addresses this by showing:
> > >
> > > > **a substantial portion of CIL performance can already be achieved using class-level statistics on frozen PTM features, even with minimal or no adaptation.**
> > >
> > > This implies that:
> > > - part of recent progress stems from stronger representations,
> > > - while downstream contributions may be overestimated without proper control.
> > >
> > > Thus, rather than diminishing CIL contributions, our work **refines how progress should be evaluated in PTM-based incremental learning**.
> > >
> > > Importantly, our goal is not to attribute gains solely to PTMs, but to show that **their role must be explicitly accounted for when assessing incremental learning methods**. Under strong PTM features, it is necessary to disentangle and reassess the true contribution of incremental strategies, rather than attributing improvements solely to new modules. We hope this perspective encourages more controlled evaluation and principled design in future CIL research.
> > >
> > > ---
> > >
> > > **W3: Failure in Complex Scenarios**
> > >
> > > We appreciate the reviewer’s observation on challenging datasets such as ImageNet-R and agree that high-overlap scenarios remain difficult.
> > >
> > > However, this reflects **an intrinsic limitation of class-level statistical modeling**, rather than a weakness of our formulation.
> > >
> > > (1) **Evidence from a training-free, full-data setting (SViT).**
> > > SViT uses all available data without parameter updates, reflecting the intrinsic capacity of statistical modeling under strong PTM features.
> > >
> > > Our modeling follows a progressive refinement:
> > > - **prototype-based modeling**,
> > > - **Gaussian modeling** (Mahalanobis distance),
> > > - **GMM** for multi-modality.
> > >
> > > Despite increased modeling complexity, performance on ImageNet-R does not improve correspondingly. Although GMM achieves higher likelihood than single Gaussian modeling, it still fails to provide sufficient discrimination.
> > >
> > > This indicates that **class-level statistical summaries are inherently insufficient to separate highly overlapping feature distributions**, rather than the issue being model flexibility.
> > >
> > > (2) **Interpretation as a boundary condition.**
> > > We view ImageNet-R as an empirical boundary where class-level statistical modeling alone becomes insufficient.
> > >
> > > This leads to a clear takeaway:
> > > - when distributions are well-separated → statistics are effective
> > > - when distributions overlap → additional mechanisms (e.g., feature adaptation or global alignment) are required
> > >
> > > In such cases, improving feature alignment is more effective than increasing probabilistic modeling complexity. We believe identifying this boundary clarifies both the strengths and limitations of statistical approaches in CIL.
> > >
> > > ---
> > >
> > > **Summary**
> > >
> > > We do not claim statistical modeling alone solves all CIL challenges. Instead, we **demonstrate its strong capability under PTM features while clearly identifying its limitations**, providing a more grounded basis for future research.

---

### Decision · Program_Chairs · 2026-04-30

**Decision:**

Accept (regular)

**Comment:**

This paper investigates the role of class-level feature statistics in CIL using pre-trained models like CLIP. IThe paper does not have a lot of technical novelty or insights, in fact the end result boils down to proposing the replacement of the CLIP score with a combination of the CLIP score and the likelihood of image features under a Gauss Mixture Model.  However, the insight that simply modeling and updating feature statistics is as (or more) effective than learning model parameters, which is the approach followed by most of the literature, is interesting.  By constructing SCLIP and SViT, the authors provides a  simplified perspective on  this research area.

Reviewers were initially split, reflecting the opinions that there is not a lot of technical novelty but the finding itself is significant. They asked a number of questions that the rebuttal addressed. After the rebuttal, three reviewers were swayed by the significance of the findings, while one remained skeptical. However, this reviewer does not seem to question  the findings or their significance but instead raises many questions about the presentation of the paper. These were not considered sufficiently serious to prevent the publication of the paper.